# No Algorithmic Collusion in Two-Player Blindfolded Games with Thompson Sampling

## Abstract

When two players are engaged in a repeated game with unknown payoff matrices, they may be completely unaware of the existence of each other and use multi-armed bandit algorithms to choose the actions, which is referred to as the "blindfolded game" in this paper. We show that when the players use Thompson sampling, the game dynamics converges to the Nash equilibrium under a mild assumption on the payoff matrices. Therefore, algorithmic collusion doesn't arise in this case despite the fact that the players do not intentionally deploy competitive strategies. To prove the convergence result, we find that the framework developed in stochastic approximation doesn't apply, because of the sporadic and infrequent updates of the inferior actions and the lack of Lipschitz continuity. We develop a novel sample-path-wise approach to show the convergence.

## 1 Introduction

Algorithmic collusion refers to the market phenomenon that when two or more competing parties use algorithms to assist decision-making, over time it may unintentionally lead to collusion instead of the typical Nash equilibrium. For example, consider two firms setting prices for their products, which are competing for customers. In the classic Bertrand competition, when the demand functions (how the market demand for either product depends on the prices of itself and the competitor) for both products are common knowledge, the firms may charge \$10 in the (symmetric) Nash equilibrium. On the other hand, when the demand functions are unknown initially, the two firms may deploy reinforcement learning algorithms to learn the demand functions and maximize profits simultaneously. Algorithmic collusion emerges when the long-term outcome of the algorithms is an equilibrium in which both firms charge more than \$10 for the products.

It has been shown in the recent literature that algorithmic collusion is possible in theoretical and experimental settings (Calvano et al., 2020; Hansen et al., 2021; Meylahn & V. den Boer, 2022). The settings of the studies usually differ in terms of the choice of algorithms and the information structure such as whether the players observe the past actions and payoffs of other players. Many studies show that all players using specifically designed algorithms (which usually requires some knowledge of the other players and sometimes synchronization among players) can lead to algorithmic collusion.

**Setup.** In this paper, on the contrary, we study a repeated game with a simple and straightforward setup, which we refer to as "*blindfolded game*." We consider two players and each player has two actions $(i, j) \in \{1, 2\}^2$. The expected payoffs for actions $(i, j)$ are $(A_{i,j}, B_{i,j})$ for the two players, respectively, although the players don't know the payoff matrix $\{(A_{i,j}, B_{i,j})\}_{i,j=1}^2$ initially. The realized payoffs of the game in round $n$ for the two players, $(a_n, b_n)$, depend on the actions taken by the two players in that round $(i_n, j_n)$. In particular, their expected values are $A_{i_n,j_n}$ and $B_{i_n,j_n}$, respectively. We consider a *zero-information* setting where the players only observe the past actions and payoffs of themselves, not their competitor's, thus referred to as blindfolded. In fact, the players don't have to be aware of the existence of the other player. We emphasize that the blindfolded game requires the least amount of information among the studies in the literature: the players only observe their own actions and realized payoffs in the past, without observing any information of the competitor or the knowledge of the payoff matrix. This resembles many real-word business settings such as price competition: the firm usually doesn't have the data on where the eroded market share is directed to, at least in the short run. In this setting, from a player's point of view, the repeated game

can be treated as stochastic multi-armed bandits (MAB) where her two actions are regarded as two arms. We investigate the scenario when both players apply Thompson sampling, a popular Bayesian algorithm in MAB and reinforcement learning (Russo et al., 2018).

**Informal results.**    We find that when the payoff matrix satisfies a mild condition and the game has a unique pure-strategy Nash equilibrium, the actions of the two players converge to the Nash equilibrium as $n \to \infty$. In other words, algorithmic collusion doesn't emerge and the supra-competitive outcomes will not arise. This is surprising: the realized payoff of a player in each round depends on the action of the competitor but Thompson sampling completely ignores such dependence. Therefore, viewed from the lens of multi-armed bandit, the payoffs are distorted and non-stationary. Still, Thompson sampling converges to the Nash equilibrium. Note that not all multi-armed bandit algorithms have this property; see a counterexample for UCB in Hansen et al. (2021). In contrast to the literature, our result demonstrates the robustness of the concept of Nash equilibrium.

**Technical contribution and the connection to the literature.**    Our approach relies on two crucial steps: First, we construct a stochastic system that represents the evolution of the blindfolded game under Thompson sampling, such as the posterior distribution of the two arms for both players. The dynamics resemble a system that can be analyzed using stochastic approximation Kushner & Yin (2003). However, a few unique features of the problem make the existing theories of stochastic approximation unable to be applied. In particular, there are three potential existing approaches. (1) Stochastic approximation with two time scales Borkar (1997) requires the state to be updated simultaneously but with different scales, while in our system, the posterior of the inferior action is only updated infrequently and sporadically. (2) Asynchronous stochastic approximation Borkar (1998) allows the states to be updated in different rounds, but the updating frequencies need to be of the same order. In contrast, in our system, because of Thompson sampling, the inferior action is only taken $O(\log n)$ of the $n$ rounds. The two challenges above make the standard framework developed in stochastic approximation, such as the analysis of the associated ODEs, unsuitable to be applied. (3) The closest study to our problem is Tsitsiklis (1994), which uses a sample-path-wise argument instead of an ODE-based approach. However, this study relies on a crucial assumption: the Lipschitz continuity of the dynamics w.r.t. the state. In our system, however, when the posterior variances are very small, the system is not globally Lipschitz continuous in the neighborhood where the empirical means of the actions/arms are equal. Therefore, in our second step, we use the sample-path-wise approach from scratch to overcome these challenges. It greatly extends the approach used in Tsitsiklis (1994). The proof strategy is novel and has not been seen in the literature before. The discussion is summarized in Table 1.

| Approach | Literature | Challenge |
|---|---|---|
| SA with two time scales | Borkar (1997; 2009) | Not updated simultaneously |
| Asynchronous SA | Borkar (1998) | Updated with very different frequencies |
| Sample-path-wise argument | Tsitsiklis (1994) | Not globally Lipschitz continuous |

Table 1: Connection to the literature on proving the convergence of the system

## 1.1 OTHER RELATED LITERATURE

Algorithmic collusion has attracted the attention of scholars and regulators recently. Calvano et al. (2020) demonstrate using simulation that when two competing firms both use Q-learning algorithms, the set prices may converge to an collusive equilibrium higher than the Nash equilibrium, although the two firms do not collude explicitly. Similar phenomena have been observed for UCB (Hansen et al., 2021) or more sophisticated learning algorithms Meylahn & V. den Boer (2022); Aouad & den Boer (2021); Klein (2021). We demonstrate a negative result: algorithmic collusion cannot arise in the blindfolded game. The key difference in our setup is the lack of information communication: the algorithms in the literature typically require the competitors' past actions or a shared state of the system as inputs. For example, in Calvano et al. (2020), each player remembers the prices of *all* players in the last $k$ rounds and uses it as a state for the Q-learning algorithm. Therefore, comparing the setups, our result supports the claim that the forced disclosure or transparency of firms in a

market may backfire and lead to algorithmic collusion. In a recent paper, (Calvano et al., 2021) show that algorithmic collusion can still emerge in low-information settings with $\epsilon$-greedy-based $Q$-learning. Hence, our result may also be specific to the nature of Thompson sampling, which encourages sufficient exploration.

Repeated games and learning have been a classic topic in economics (Fudenberg et al., 1998). The convergence of fictitious play has been studied extensively Hofbauer & Sandholm (2002). Besides fictitious play, Cesa-Bianchi & Lugosi (2006) provide a summary of classic results: if all players adopt no-regret algorithms (sublinear regret against adversaries), then the empirical distribution of the actions converges to the coarse correlated Nash equilibrium. Since then, there has been a growing body of literature on multi-agent learning in games. The focus has been shifted toward the so-called last-iterate convergence instead of the empirical distribution (Mertikopoulos & Staudigl, 2017; Mertikopoulos & Zhou, 2019; Mazumdar et al., 2020) Perkins et al. (2015). A survey can be found in Yang & Wang (2020). This study also focuses on the last-iterate convergence. The main difference of our setup is that the actions are not continuous and the players do not receive first-order feedback. This setup is first proposed in Ortega & Braun (2014) and the convergence is shown numerically. O'Donoghue et al. (2021) show that using Thompson sampling in games when the competitor plays a different policy can lead to linear regret. In contrast, in our setup, both players use Thompson sampling.

Our study deviates from multi-agent reinforcement learning in terms of the motivation and research question. In multi-agent reinforcement learning (Zhang et al., 2021; Yang et al., 2018) or multi-agent Thompson sampling (Verstraeten et al., 2020), the goal is to design algorithms and communication protocols that only rely on the local information of each agent to achieve convergence to the co-operative or Nash equilibrium. In our study, we do not design new algorithms but document the dynamics under the classic Thompson sampling. There is no communication between the players either. Thompson sampling has been a popular algorithm for stochastic multi-armed bandit. A tutorial of Thompson sampling is given in Russo et al. (2018) and the theoretical property is proved in, e.g., Kaufmann et al. (2012); Agrawal & Goyal (2012). The introduction of the MAB setup and other algorithms can be found in Bubeck & Cesa-Bianchi (2012); Lattimore & Szepesvári (2020).

To conclude this introduction, we mention some additional studies on stochastic approximation. While we focus on asymptotic convergence analysis, we note that there is a growing body of literature recently on finite-time analysis of SA, see, e.g., Srikant & Ying (2019); Qu & Wierman (2020); Chen et al. (2021); Haque et al. (2023) and references therein.

## 2 TWO-PLAYER BLINDFOLDED GAME WITH THOMPSON SAMPLING

### 2.1 PROBLEM FORMULATION

Consider a game with two players, player 1 and player 2, each having 2 possible actions $\{1, 2\}$. The payoff of the game is represented by $G = (A, B)$, where $A, B$ are both $2 \times 2$ matrices. In particular, the expected payoff of player 1 is $A_{i,j}$ where $i, j \in \{1, 2\}$ are the actions taken by player 1 and player 2, respectively. Similarly $B_{i,j}$ is the expected payoff of player 2 under the same action profile. The game is played repeatedly. We use $i_n$ and $j_n$ to denote the actions taken by player 1 and player 2 in round $n$. Given $i_n$ and $j_n$, the *realized* payoffs in round $n$ are normal random variables: $a_{i_n,n} \sim \mathcal{N}(A_{i_n,j_n}, 1)$ and $b_{j_n,n} \sim \mathcal{N}(B_{i_n,j_n}, 1)$, where $\mathcal{N}(\mu, \sigma^2)$ denotes the normal distribution with mean $\mu$ and variance $\sigma^2$.

We consider a specific strategy for both players. In particular, both players treat the two actions as two arms, and use Thompson sampling (Russo et al., 2018) ignoring the existence of the other player. Roughly speaking, Thompson sampling assumes a prior distribution for the unknown mean of the two arms. At every time step, play an arm according to its posterior probability of being the best arm. We refer to this as the *blindfolded game*, as if the players are not aware of the game and simply conduct stochastic multi-armed bandits. We formally state the dynamics of the game in Algorithm 1.

In the blindfolded game, both players cannot (or do not need to) observe the past actions and payoffs of the other player. They only keep track of the past actions and payoffs of themselves. It is arguably the most realistic setting in business, when algorithmic collusion attracts the attention of regulators. The firms usually don't realize and react to the competitive pressure from new entrants. Even if they

---

**Algorithm 1** Two-Player Blindfolded Game with Thompson Sampling

---

**Require:** Payoff matrices $G = (A, B)$

1: Initialize: number of pulls for both actions for player 1, $N_{i,0} = 0$ $(i = 1, 2)$, and for player 2, $M_{j,0} = 0$ $(j = 1, 2)$

2: **for** $n = 1, 2, \ldots$ **do**

3:  player 1: for action $k = 1, 2$, sample $\theta_{k,n}$ independently from Gaussian distribution $\mathcal{N}\left(\frac{\sum_{s=1}^{n-1} a_{i_s,s} \cdot \mathbf{1}_{\{i_s=k\}}}{N_{k,n-1}+1}, \frac{1}{N_{k,n-1}+1}\right)$ where $N_{k,n-1} = \sum_{s=1}^{n-1} \mathbf{1}_{\{i_s=k\}}$, then choose the action $i_n = \arg\max_k \theta_{k,n}$.

4:  player 2: for action $k = 1, 2$, sample $\rho_{k,n}$ independently from Gaussian distribution $\mathcal{N}\left(\frac{\sum_{s=1}^{n-1} b_{j_s,s} \cdot \mathbf{1}_{\{j_s=k\}}}{M_{k,n-1}+1}, \frac{1}{M_{k,n-1}+1}\right)$ where $M_{k,n-1} = \sum_{s=1}^{n-1} \mathbf{1}_{\{j_s=k\}}$, then choose the action $j_n = \arg\max_k \rho_{k,n}$.

5:  Observe the reward $a_{i_n,n} \sim \mathcal{N}(A_{i_n,j_n}, 1)$ and $b_{j_n,n} \sim \mathcal{N}(B_{i_n,j_n}, 1)$ for two players.

6: **end for**

---

do, the past actions or payoffs of the competitor are typically confidential. It is reasonable to assume that the firms just focus on the business decisions of their own, and deploy single-agent reinforcement learning algorithms, among which Thompson sampling is probably the simplest one.

Note that Thompson sampling is *not correctly specified*. When considering the actions of the other player, the expected payoffs of both arms/actions are not stationary. Moreover, although the two players are blindfolded, their actions are tightly coupled through the realized payoffs they observe, which feed into the posterior distributions in a highly nonlinear way. Therefore, a priori it is not clear how the game evolves or whether it converges. In the rest of the paper, we will show that, surprisingly, the game converges to the Nash equilibrium under a set of general conditions. As a result, there is no algorithmic collusion in two-player blindfolded games with Thompson sampling.

## 2.2 GAME DYNAMICS

We first introduce a number of states to record the system dynamics for the two-player blindfolded game with Thompson sampling. For player 1, we define for $i = 1, 2$,

$$x_{i,n} := \begin{cases} 0, & \text{if } n = 0, \\ \frac{\sum_{s=1}^n a_{i_s,s} \cdot \mathbf{1}_{\{i_s=i\}}}{N_{i,n}+1}, & \text{if } n \geq 1, \end{cases} \quad \text{and} \quad w_{i,n} := \begin{cases} 1, & \text{if } n = 0, \\ \frac{1}{N_{i,n}+1}, & \text{if } n \geq 1, \end{cases} \tag{1}$$

where $N_{i,n} := \sum_{s=1}^n \mathbf{1}_{\{i_s=i\}}$ denotes the number of plays of action $i$ by Player 1 up to round $n$. It is clear that $x_{i,n}$ is the empirical mean of action $i$ after $n$ rounds. For Thompson sampling with Gaussian priors and unit-variance Gaussian reward observations, $x_{i,n}$ and $w_{i,n}$ represent the mean and variance of the posterior Gaussian distribution at the beginning of round $n + 1$ of action $i$ for player 1, see, e.g., Russo et al. (2018). Similarly, we define for player 2, for $j = 1, 2$,

$$y_{j,n} := \begin{cases} 0, & \text{if } n = 0, \\ \frac{\sum_{s=1}^n b_{j_s,s} \cdot \mathbf{1}_{\{j_s=j\}}}{M_{j,n}+1}, & \text{if } n \geq 1, \end{cases} \quad \text{and} \quad v_{j,n} := \begin{cases} 1, & \text{if } n = 0, \\ \frac{1}{M_{j,n}+1}, & \text{if } n \geq 1, \end{cases} \tag{2}$$

with $M_{j,n} := \sum_{s=1}^n \mathbf{1}_{\{j_s=j\}}$ denoting the number of plays of action $j$ by Player 2 up to round $n$. Then the system state for the blindfolded game is denoted by $\boldsymbol{S}_n$ at time $n$, which is defined by

$$\boldsymbol{S}_n := (x_{1,n}, x_{2,n}, y_{1,n}, y_{2,n}, w_{1,n}, w_{2,n}, v_{1,n}, v_{2,n}) \in \mathbb{R}^4 \times \mathbb{R}_+^4, \tag{3}$$

where $\mathbb{R}_+ = (0, \infty)$. Note that $\boldsymbol{S}_n$ is a sufficient statistics for both players to sample their actions in round $n + 1$ based on Algorithm 1.

We next discuss the dynamics of the state $\boldsymbol{S}_n$. We focus on the dynamics of $x_{i,n}$ and $w_{i,n}$ for $i = 1, 2$. By symmetry, we can express the dynamics of the other states similarly. For player 1, if action $i \in \{1, 2\}$ is chosen in round $n + 1$, then $N_{i,n+1} = N_{i,n} + 1$ and $N_{-i,n+1} = N_{-i}$ where $-i$ is the

action other than $i$. We can infer from (1) that

$$x_{i,n+1} = \frac{\sum_{s=1}^{n+1} a_{i_s,s} \cdot \mathbf{1}_{\{i_s=i\}}}{N_{i,n+1}+1} = \frac{x_{i,n}(N_{i,n}+1) + a_{i,n+1}}{N_{i,n+1}+1} = x_{i,n} + \frac{-x_{i,n} + a_{i,n+1}}{N_{i,n+1}+1},$$

$$w_{i,n+1} = \frac{1}{N_{i,n+1}+1} = \frac{w_{i,n}(N_{i,n}+1)}{N_{i,n+1}+1} = w_{i,n} + \frac{-w_{i,n}}{N_{i,n+1}+1}.$$

If action $i$ is not chosen at round $n+1$, then $N_{i,n+1} = N_{i,n}$, and it is easy to see that

$$x_{i,n+1} = x_{i,n}, \quad w_{i,n+1} = w_{i,n}.$$

Combining these two cases, we obtain for $i = 1, 2$,

$$x_{i,n+1} = x_{i,n} + \alpha_{i,n+1} \cdot (-x_{i,n} + a_{i,n+1}), \quad \text{and} \quad w_{i,n+1} = w_{i,n} + \alpha_{i,n+1} \cdot (-w_{i,n}), \quad (4)$$

where $\alpha_{i,n+1}$ are binary-valued random variables with $\alpha_{i,n+1} = \frac{1}{N_{i,n+1}+1}$ if action $i$ is selected in round $n+1$ and $\alpha_{i,n+1} = 0$ otherwise.

To express the dynamics (4) in terms of the current state $\boldsymbol{S}_n$, we need to understand the probability distribution of $\alpha_{i,n+1}$ and $a_{i,n+1}$. Note that with Thompson sampling, given the information $\mathcal{F}_n$ up to round $n$, the probability that player 1 chooses action $i$ in round $n+1$ is given by

$$\varphi_{i,n+1} := \mathsf{P}(i_{n+1} = i|\mathcal{F}_n) = \mathsf{P}(\theta_{i,n+1} \geq \theta_{-i,n+1}|\boldsymbol{S}_n) = \Phi\left(\frac{x_{i,n} - x_{-i,n}}{\sqrt{w_{i,n} + w_{-i,n}}}\right), \quad (5)$$

where recall that $\theta_{i,n+1} \sim \mathcal{N}(x_{i,n}, w_{i,n})$, and $\Phi(\cdot)$ denotes the cumulative distribution function of a standard normal distribution. This implies that for $i = 1, 2$,

$$\mathsf{P}\left(\alpha_{i,n+1} = \frac{1}{N_{i,n+1}+1}\bigg|\boldsymbol{S}_n\right) = \varphi_{i,n+1}, \quad \text{and} \quad \mathsf{P}(\alpha_{i,n+1} = 0|\boldsymbol{S}_n) = 1 - \varphi_{i,n+1}. \quad (6)$$

Next we analyze the term $a_{i,n+1}$. When player 1 selects action $i \in \{1, 2\}$ and player 2 selects action $j_{n+1}$ in round $n+1$, the reward is $a_{i,n+1} \sim \mathcal{N}(A_{i,j_{n+1}}, 1)$ (see Algorithm 1). We can rewrite the expression as $a_{i,n+1} = \sum_{j=1}^{2} A_{i,j} \mathbf{1}_{\{j_{n+1}=j\}} + \epsilon_{i,n+1}$, where $\epsilon_{i,n+1} \sim \mathcal{N}(0, 1)$ is the noise independent of everything else. Our goal is to decompose $a_{i,n+1}$ into a term that is adapted to $\mathcal{F}_n$ and a martingale-difference term. To do so, denote by $\psi_{j,n+1}$ the probability that player 2 selects action $j \in \{1, 2\}$ in round $n+1$. It is given by

$$\psi_{j,n+1} := \mathsf{P}(j_{n+1} = j|\mathcal{F}_n) = \mathsf{P}(\rho_{j,n+1} \geq \rho_{-j,n+1}|\boldsymbol{S}_n) = \Phi\left(\frac{y_{j,n} - y_{-j,n}}{\sqrt{v_{j,n} + v_{-j,n}}}\right), \quad (7)$$

where recall that $\rho_{j,n+1} \sim \mathcal{N}(y_{j,n}, v_{j,n})$ by Algorithm 1. Hence, given action $i$ is chosen by player 1 in round $n+1$, we have $a_{i,n+1} = \sum_j A_{i,j} \psi_{j,n+1} + \bar{a}_{i,n+1}$, where

$$\bar{a}_{i,n+1} = \left[\sum_{j=1}^{2} A_{i,j} \mathbf{1}_{\{j_{n+1}=j\}} - \sum_{j=1}^{2} A_{i,j} \psi_{j,n+1}\right] + \epsilon_{i,n+1}. \quad (8)$$

It is easy to see that $\bar{a}_{i,n+1}$ has mean zero conditional on $\mathcal{F}_n$ or $\boldsymbol{S}_n$. This allows us to rewrite (4) as follows:

$$x_{i,n+1} = x_{i,n} + \alpha_{i,n+1} \cdot (-x_{i,n} + \sum_j A_{i,j} \psi_{j,n+1} + \bar{a}_{i,n+1}), \quad w_{i,n+1} = w_{i,n} + \alpha_{i,n+1} \cdot (-w_{i,n}). (9)$$

Analogously, we can derive for $j = 1, 2$,

$$y_{j,n+1} = y_{j,n} + \beta_{j,n+1} \cdot (-y_{j,n} + \sum_{i=1}^{2} B_{i,j} \varphi_{i,n+1} + \bar{b}_{j,n+1}), \quad v_{j,n+1} = v_{j,n} + \beta_{j,n+1} \cdot (-v_{j,n}). (10)$$

Here, $\beta_{j,n+1}, j = 1, 2$, are also binary-valued random variables with

$$\mathsf{P}\left(\beta_{j,n+1} = \frac{1}{M_{j,n+1}+1}\bigg|\boldsymbol{S}_n\right) = \psi_{j,n+1}, \quad \text{and} \quad \mathsf{P}(\beta_{j,n+1} = 0|\boldsymbol{S}_n) = 1 - \psi_{j,n+1}, \quad (11)$$

and

$$\bar{b}_{j,n+1} = \left[\sum_{i=1}^{2} B_{i,j}\mathbf{1}_{\{i_{n+1}=i\}} - \sum_{i=1}^{2} B_{i,j}\varphi_{i,n+1}\right] + \tilde{\epsilon}_{j,n+1}, \tag{12}$$

where $(\tilde{\epsilon}_{i,n})$ are i.i.d standard normal random noise independent of everything else.

In a vector form, given $\boldsymbol{S} := (x_1, x_2, y_1, y_2, w_1, w_2, v_1, v_2) \in \mathbb{R}^4 \times \mathbb{R}_+^4$, we define a function $F(\cdot) : \mathbb{R}^4 \times \mathbb{R}_+^4 \mapsto \mathbb{R}^8$ as follows:

$$F(\boldsymbol{S}) := (A_{1,1}\psi_1 + A_{1,2}\psi_2, A_{2,1}\psi_1 + A_{2,2}\psi_2, B_{1,1}\varphi_1 + B_{2,1}\varphi_2, B_{1,2}\varphi_1 + B_{2,2}\varphi_2, 0, 0, 0, 0), \tag{13}$$

where

$$\varphi_1 = \Phi\left(\frac{x_1 - x_2}{\sqrt{w_1 + w_2}}\right) = 1 - \varphi_2 \quad \text{and} \quad \psi_1 = \Phi\left(\frac{y_1 - y_2}{\sqrt{v_1 + v_2}}\right) = 1 - \psi_2. \tag{14}$$

Then we can vectorize (9) and (10) that the dynamics of $\boldsymbol{S}_n$ in (3) is given by

$$\boldsymbol{S}_{n+1} - \boldsymbol{S}_n = \boldsymbol{\gamma}_{n+1} \circ \left(F(\boldsymbol{S}_n) - \boldsymbol{S}_n + \bar{\boldsymbol{\xi}}_{n+1}\right), \tag{15}$$

where $\boldsymbol{\gamma}_{n+1} := (\alpha_{1,n+1}, \alpha_{2,n+1}, \beta_{1,n+1}, \beta_{2,n+1}, \alpha_{1,n+1}, \alpha_{2,n+1}, \beta_{1,n+1}, \beta_{2,n+1})$, the notation $\circ$ denotes the component-wise multiplication, and

$$\bar{\boldsymbol{\xi}}_{n+1} := (\bar{a}_{1,n+1}, \bar{a}_{2,n+1}, \bar{b}_{1,n+1}, \bar{b}_{2,n+1}, 0, 0, 0, 0), \tag{16}$$

where $\bar{a}_{i,n+1}$ and $\bar{b}_{j,n+1}$ are given in (8) and (12) respectively.

From (15), it is clear that the system dynamics can be described by a special form of *stochastic approximation* (Kushner & Yin, 2003). In particular, $\boldsymbol{\gamma}_{n+1}$ is the random (vectorized) "step size" and the term $\bar{\boldsymbol{\xi}}_{n+1}$ is the noise conditional on $\boldsymbol{S}_n$.

## 2.3 EQUILIBRIUM POINT OF THE GAME

Next we impose assumptions on the game itself for our theoretical analysis of the system.

**Assumption 1.** *(1) There are no ties in the payoffs: $A_{i,j} \neq A_{i',j}$ and $B_{i,j} \neq B_{i,j'}$ for $i \neq i', j \neq j'$. (2) There is a unique pure-strategy Nash equilibrium.*

The first part of the assumption has also appeared in the literature Wunder et al. (2010). For the second part, by symmetry, assume $(1, 1)$ is the unique pure-strategy Nash equilibrium. It implies that one of following three cases hold

Case 1. $A_{1,1} > A_{2,1}, A_{1,2} > A_{2,2}, B_{1,1} > B_{1,2}, B_{2,1} > B_{2,2}$.

Case 2. $A_{1,1} > A_{2,1}, A_{1,2} > A_{2,2}, B_{1,1} > B_{1,2}, B_{2,1} < B_{2,2}$.

Case 3. $A_{1,1} > A_{2,1}, A_{1,2} < A_{2,2}, B_{1,1} > B_{1,2}, B_{2,1} > B_{2,2}$.

For the two-player two-action game we consider, Vega-Redondo (2003) point out that there are two additional cases: there may be a unique mixed-strategy Nash equilibrium, or there may be two pure-strategy Nash equilibria and one mixed-strategy Nash equilibrium. In this study, we mainly focus on the game satisfying Assumption 1 for analytical tractability. In particular, we can show that there is a unique equilibrium point that the system (14) may converge to corresponding to the pure-strategy Nash equilibrium $(1, 1)$:

$$\boldsymbol{S}^* = (x_1^*, x_2^*, y_1^*, y_2^*, w_1^*, w_2^*, v_1^*, v_2^*) = (A_{1,1}, A_{2,1}, B_{1,1}, B_{1,2}, 0, 0, 0, 0). \tag{17}$$

The rest of the paper develop an approach to prove the almost sure convergence of $\boldsymbol{S}_n$ to $\boldsymbol{S}^*$.

## 3 PRELIMINARY RESULTS

To establish the convergence of $\boldsymbol{S}_n$, we first show the following results. The proofs are given in Appendix A.2, A.3, A.4 and A.5.

**Lemma 1.** $\lim_{n\to\infty} N_{i,n} = \infty$ *almost surely for* $i = 1, 2$. *Similarly,* $\lim_{n\to\infty} M_{j,n} = \infty$ *almost surely for* $j = 1, 2$.

Lemma 1 states that for each player, both actions have been taken infinitely often. Recall the random step sizes $(\alpha_{i,n})_{i=1,2}$ and $(\beta_{j,n})_{j=1,2}$ given in (6) and (11), and the noise $\bar{\boldsymbol{\xi}}_n$ given in (16).

**Lemma 2.** *For* $i = 1, 2$, $\sum_{n=1}^{+\infty} \alpha_{i,n} = \infty$ *and* $\sum_{n=1}^{+\infty} \alpha_{i,n}^2 < \infty$ *almost surely. Similarly,* $\sum_{n=1}^{+\infty} \beta_{j,n} = \infty$ *and* $\sum_{n=1}^{+\infty} \beta_{j,n}^2 < \infty$ *almost surely for* $j = 1, 2$.

Note that the nature of Lemmas 1 and 2 is different from the stochastic MAB, because the exploration of actions also depends on the past actions of the other player. Using the mechanism of Thompson sampling, we show that the past actions of the other player do not hinder the exploration. It is also important to note that unlike in standard SA, the step sizes $(\alpha_{i,n})$ and $(\beta_{j,n})$ in our study are randomly sampled (6) and are not standard decreasing sequences (e.g. $\alpha_{i,n} = 0$ is arm $i$ is not pulled in round $n$).

We also have the following two results.

**Lemma 3** (Martingale difference noise). $(\bar{\boldsymbol{\xi}}_n : n \geq 1)$ *is a martingale difference sequence with* $\mathbb{E}[\bar{\boldsymbol{\xi}}_{n+1}|\mathcal{F}_n] = 0$ *for all* $n$. *In addition, there exists* $C \geq 0$ *such that* $\mathbb{E}[\bar{\boldsymbol{\xi}}_{n+1}^2|\mathcal{F}_n] \leq C$ *for all* $n$.

**Lemma 4** (Boundedness of the iterates). $\sup_n \|\boldsymbol{S}_n\| < \infty$ *almost surely.*

The four lemmas are commonly seen in stochastic approximation. While we need them in our proof as well, as we shall see next, the proof deviates significantly from the stochastic approximation literature.

## 4 MAIN RESULT AND ANALYSIS

This section presents our main theoretical results on the convergence. Without loss of generality, suppose the game's pure-strategy Nash Equilibrium is $(1, 1)$ as in Section 2.3, i.e., both players 1 and 2 play action 1. We first state a somewhat artificial assumption that is crucial in proving the main convergence result.

**Assumption 2.** *The payoff matrices* $(A, B)$ *satisfy* $|A_{1,2} - A_{1,1}| + |A_{2,2} - A_{2,1}| < A_{1,1} - A_{2,1}$ *and* $|B_{2,1} - B_{1,1}| + |B_{2,2} - B_{1,2}| < B_{1,1} - B_{1,2}$.

The assumption states that the payoff of the Nash equilibrium cannot be much worse than the other actions. It plays an instrumental role in our sample-path-wise argument. It remains unknown theoretically if the convergence can be guaranteed without this assumption, although numerical experiments indicate that convergence can still be achieved. Next we state the main result.

**Theorem 1.** *Suppose Assumptions 1 and 2 hold. The state that encodes the game dynamics* $\boldsymbol{S}_n$ *in* (3) *converges to* $\boldsymbol{S}^*$ *almost surely as* $n \to \infty$, *where* $\boldsymbol{S}^*$ *is the equilibrium point.*

Theorem 1 implies that $x_{i,n} \to x_i^*$ almost surely. That is, the average payoffs of playing action $i$ for player one converge to $A_{i,1}$. Similarly, the average payoffs of playing action $j$ for player two converge to $B_{1,j}$.

On the other hand, from the three cases after Assumption 1 and the equilibrium point (17), we can see that $x_1^* = A_{1,1} > A_{2,1} = x_2^*$ and $y_1^* = B_{1,1} > B_{1,2} = y_2^*$. Therefore, in the limit, the probability of playing action 1 by player 1 converges to

$$\lim_{n\to\infty} \varphi_{1,n+1} = \lim_{n\to\infty} \Phi\left(\frac{x_{1,n} - x_{2,n}}{\sqrt{w_{1,n} + w_{2,n}}}\right) = \Phi(\infty) = 1, \quad \text{and} \quad \lim_{n\to\infty} \varphi_{2,n+1} = 0, \quad (18)$$

where we recall that $\varphi_{i,n+1}$ denotes the probability that player 1 chooses action $i \in \{1, 2\}$ at time $n + 1$ given the information up to time $n$. Similarly, we can obtain from (7) that

$$\lim_{n\to\infty} \psi_{1,n+1} = \lim_{n\to\infty} \Phi\left(\frac{y_{1,n} - y_{2,n}}{\sqrt{v_{1,n} + v_{2,n}}}\right) = \Phi(\infty) = 1, \quad \text{and} \quad \lim_{n\to\infty} \psi_{2,n+1} = 0, \quad (19)$$

where $\psi_{j,n+1}$ denotes the probability that player 2 chooses action $j \in \{1, 2\}$ at time $n + 1$. Therefore, we deduce from (18) and (19) that the actions of the two players converge to the unique pure-strategy

Nash equilibrium as $n \to \infty$. This is referred to as the last-iterate convergence in the literature (Lin et al., 2020; Golowich et al., 2020), which is stronger than the convergence of the empirical distribution of plays.

The proof of Theorem 1 builds on the proof of Theorem 3 in (Tsitsiklis, 1994), but it is substantially more involved. Theorem 3 in (Tsitsiklis, 1994) requires the iteration mapping $F$ to be a contraction which is violated in our case. In particular, $F$ in (13) cannot be a contraction in the whole domain of $\boldsymbol{S}_n$. It is clear that when $w$ and $v$ are small, i.e., both actions of both players have been taken many times, $\varphi$ and $\psi$ are not Lipschitz continuous in the neighborhood of $x_1 = x_2$ and $y_1 = y_2$. For instance, we can easily compute that $\left| \frac{\partial \varphi_1}{\partial x_1} \right| = \frac{1}{\sqrt{2\pi}} \exp \left[ -\frac{1}{2} \left( \frac{x_1 - x_2}{\sqrt{w_1 + w_2}} \right)^2 \right] \frac{1}{\sqrt{w_1 + w_2}}$, which will blow up if $x_1 = x_2$ and $w_1 + w_2$ approaches zero.

Therefore, the intuition of the proof is to first argue that $\boldsymbol{S}_n$ will avoid the neighborhoods almost surely when $n$ tends to infinity. This is why Assumption 1 is essential. It guarantees that the equilibrium point $\boldsymbol{S}^*$ is bounded away from the neighborhood of $x_1 = x_2$ or $y_1 = y_2$.

## 5 SKETCHED PROOF OF THEOREM 1

We prove Theorem 1 by a sample-path-wise approach. See Appendix A.6 for the complete proof.

*Step 1. Show $\boldsymbol{S}_n$ will avoid the region where $F$ is not Lipschitz continuous for a sufficiently large $n$.* For $n \geq 1$, we first rewrite the dynamics (15) to

$$\boldsymbol{S}_n - \boldsymbol{S}^* = (1 - \boldsymbol{\gamma}_n) \circ (\boldsymbol{S}_{n-1} - \boldsymbol{S}^*) + \boldsymbol{\gamma}_n \circ \left( F(\boldsymbol{S}_{n-1}) - \boldsymbol{S}^* + \bar{\boldsymbol{\xi}}_n \right),$$

where $\boldsymbol{S}^*$ is defined in (17). One can verify that it can be written in a component-wise recursive form for $k = 1, \ldots, 8$: (see Lemma 5 in Appendix)

$$S_{k,n} - S_k^* = (S_{k,0} - S_k^*) \cdot \prod_{\tau=1}^{n}(1 - \gamma_{k,\tau}) + \sum_{\tau=1}^{n} \left[ \prod_{s=\tau+1}^{n} (1 - \gamma_{k,s}) \right] \gamma_{k,\tau} \left( F_k(\boldsymbol{S}_{\tau-1}) - S_k^* + \bar{\xi}_{k,\tau} \right),$$

where $S_{k,n}$ and $\gamma_{k,n}$ are the $k$-th entry of $\boldsymbol{S}_n$ and $\boldsymbol{\gamma}_n$ respectively. Hence we obtain

$$S_{k,n} - S_k^* = C_{k,n} + D_{k,n} + E_{k,n}, \tag{20}$$

where $C_{k,n} := (S_{k,0} - S_k^*) \cdot \prod_{\tau=1}^{n}(1 - \gamma_{k,\tau})$, $D_{k,n} := \sum_{\tau=1}^{n} \left[ \prod_{s=\tau+1}^{n}(1 - \gamma_{k,s}) \right] \gamma_{k,\tau}(F_k(\boldsymbol{S}_{\tau-1}) - S_k^*)$ and $E_{k,n} := \sum_{\tau=1}^{n} \left[ \prod_{s=\tau+1}^{n}(1 - \gamma_{k,s}) \right] \gamma_{k,\tau} \bar{\xi}_{k,\tau}$.

The first term $C_{k,n}$ on the right-hand-side (RHS) of (20) converges to zero as $n \to \infty$. This is because $\prod_{\tau=1}^{\infty}(1 - \gamma_{k,\tau}) = 0$ almost surely; see Lemma 7 in Appendix.

For the third term on the RHS of (20), we have $E_{k,n} = 0$, $k = 5, 6, 7, 8$ by the definition of the noise $\bar{\xi}_{k,n}$. Moreover, we obtain $\lim_{n \to \infty} E_{k,n} = 0$, $k = 1, 2, 3, 4$ from Lemma 2 of Tsitsiklis (1994).

Finally, for the second term on the RHS of (20), we can show that $|D_{k,n}| \leq |A_{k,2} - A_{k,1}|$ for $k = 1, 2$, $|D_{k,n}| \leq |B_{2,k-2} - B_{1,k-2}|$ for $k = 3, 4$, and $D_{k,n} = 0$ for $k = 5, 6, 7, 8$, for all $n$. See (30) and (31) in Appendix A.6.

On combining these three terms and using the definition of $\boldsymbol{S}^*$ in (17), we infer that for $\epsilon, \epsilon' > 0$, $|x_{i,n} - A_{i,1}| \leq |A_{i,2} - A_{i,1}| + \epsilon$ for $i = 1, 2$, and $|y_{j,n} - B_{1,j}| \leq |B_{2,j} - B_{1,j}| + \epsilon'$ for $j = 1, 2$, when $n \geq N_0$ for some large $N_0$. Then by Assumption 2, we obtain for $n \geq N_0$,

$$x_{1,n} - x_{2,n} > \frac{\epsilon_1}{2}(A_{1,1} - A_{2,1}), \quad y_{1,n} - y_{2,n} > \frac{\epsilon_2}{2}(B_{1,1} - B_{1,2}),$$

for some small $\epsilon_1, \epsilon_2 > 0$. It follows that that $\boldsymbol{S}_n$ will avoid the region where $F$ is not Lipschitz continuous for a sufficiently large $n$. Note that $N_0$ is random and it depends on the sample path.

*Step 2. Prove the Lipschitz constant of $F$ is smaller than 1.* More precisely, we show there exists $\eta_0 > N_0$ and $\delta \in [0, 1)$ such that for $n > \eta_0$,

$$\|F(\boldsymbol{S}_n) - F(\boldsymbol{S}^*)\|_\infty \leq \delta \|\boldsymbol{S}_n - \boldsymbol{S}^*\|_\infty.$$

To prove the result, we first apply the mean value theorem and obtain for $i = 1, \ldots, 4$,

$$F_i(\boldsymbol{S}_n) - F_i(\boldsymbol{S}^*) = \nabla F_i(\tilde{\boldsymbol{S}}_n) \cdot (\boldsymbol{S}_n - \boldsymbol{S}^*),$$

where $\tilde{\boldsymbol{S}}_n = (\tilde{x}_{1,n}, \tilde{x}_{2,n}, \tilde{y}_{1,n}, \tilde{y}_{2,n}, \tilde{w}_{1,n}, \tilde{w}_{2,n}, \tilde{v}_{1,n}, \tilde{v}_{2,n})$ is a point on the segment between $\boldsymbol{S}_n$ and $\boldsymbol{S}^*$. Hence it suffices to bound the gradient $\nabla F_i(\tilde{\boldsymbol{S}}_n)$. Denote by $L_{i,j}^n = \frac{\partial F_i(\boldsymbol{S})}{\partial S_j}|_{\boldsymbol{S} = \tilde{\boldsymbol{S}}_n}, i = 1, \ldots, 4, j = 1, \ldots, 8$. We analyze $i = 1$ for illustration. We can calculate

$$L_{1,1}^n = L_{1,2}^n = L_{1,5}^n = L_{1,6}^n = 0,$$

$$|L_{1,3}^n| = |L_{1,4}^n| = \frac{|A_{1,1} - A_{1,2}|}{\sqrt{2\pi}} \cdot e^{-\frac{1}{2} z_{1,n}^2} \cdot z_{1,n} \cdot \frac{1}{\tilde{y}_{1,n} - \tilde{y}_{2,n}},$$

$$|L_{1,7}^n| = |L_{1,8}^n| = \frac{|A_{1,1} - A_{1,2}|}{2\sqrt{2\pi}} \cdot e^{-\frac{1}{2} z_{1,n}^2} \cdot z_{1,n}^3 \cdot \frac{1}{(\tilde{y}_{1,n} - \tilde{y}_{2,n})^2},$$

where $z_{1,n} := \frac{\tilde{y}_{1,n} - \tilde{y}_{2,n}}{\sqrt{\tilde{v}_{1,n} + \tilde{v}_{2,n}}}$, $z_{2,n} := \frac{\tilde{x}_{1,n} - \tilde{x}_{2,n}}{\sqrt{\tilde{w}_{1,n} + \tilde{w}_{2,n}}}$. By Step 1 and the definition of $\tilde{\boldsymbol{S}}_n$, we have $\tilde{x}_{1,n} - \tilde{x}_{2,n} > \frac{\epsilon_1}{2}(A_{1,1} - A_{2,1}), \tilde{y}_{1,n} - \tilde{y}_{2,n} > \frac{\epsilon_2}{2}(B_{1,1} - B_{1,2}), \tilde{w}_{1,n} + \tilde{w}_{2,n} \le w_{1,n} + w_{2,n} = \frac{1}{N_{1,n}+1} + \frac{1}{N_{2,n}+1}, \tilde{v}_{1,n} + \tilde{v}_{2,n} \le v_{1,n} + v_{2,n} = \frac{1}{M_{1,n}+1} + \frac{1}{M_{2,n}+1}$. From Lemma 1, we have $\lim_{n \to \infty} N_{j,n} = \infty$ $(i = 1, 2)$ and $\lim_{n \to \infty} M_{j,n} = \infty$ $(j = 1, 2)$ almost surely. Thus, we can prove that there exists $\eta_0$ such that $0 \le L_{i,j}^n < 1$ for $n > \eta_0, i = 1, \ldots, 4, j = 1, \ldots, 8$.

*Step 3. Obtain the convergence of $\boldsymbol{S}_n$ to $\boldsymbol{S}^*$*. On combining the above two steps and applying Theorem 3 in Tsitsiklis (1994), we have $\boldsymbol{S}_n$ converging to $\boldsymbol{S}^*$.

# 6 SIMULATION STUDIES

In this section, we present results from simulation studies. The experiments are conducted on a PC with 2.10 GHz Intel Processor and 16 GB of RAM. We first consider a game that satisfies Assumptions 1 and 2 and verify our theoretical prediction. The payoff matrices are $A_1 = \begin{pmatrix} 0.5 & 0.4 \\ 0.2 & 0.3 \end{pmatrix}$, $B_1 = \begin{pmatrix} 0.7 & 0.3 \\ 0.6 & 0.5 \end{pmatrix}$ and the unique pure-strategy Nash equilibrium is $(1, 1)$. We simulate two sets of games with different prior distributions for the reward of the actions.

Case 1. Both players have prior distributions $\mathcal{N}(0, 1)$ for both actions.

Case 2. Player 1 has prior distributions $\mathcal{N}(0.2, 1)$ for action 1 and $\mathcal{N}(0.6, 1)$ for action 2, while player 2 has prior distributions $\mathcal{N}(0.4, 1)$ for action 1 and $\mathcal{N}(0.5, 1)$ for action 2.

Case 2 is designed to check if the game can converge to the Nash equilibrium when the prior distributions favor the action not in the equilibrium. We plot the probability that each player chooses action 1 for a random sample path ($x$-axis is in logarithmic scale) from round 1 to $3 \times 10^5$. The solid (dashed) curves correspond to case one (two). From Figure 2a, the two solid curves tell us that the game converges to the Nash equilibrium (probabilities converge to 1) which can verify the convergence result in Theorem 1. Besides, the two dashed curves show that although players start from the incorrect prior distribution of each action, the game will still converge to the Nash equilibrium.

We then consider a game that satisfies Assumption 1 while Assumption 2 is violated, whose payoff matrices are $A_2 = \begin{pmatrix} 0.2 & 0.5 \\ 0.1 & 0.4 \end{pmatrix}$, $B_2 = \begin{pmatrix} 0.2 & 0.1 \\ 0.5 & 0.4 \end{pmatrix}$ with the unique pure-strategy Nash equilibrium $(1, 1)$. This game corresponds to the prisoner's dilemma. We find from Figure 2b that the game will still converge to the Nash equilibrium with the above two prior distributions cases.

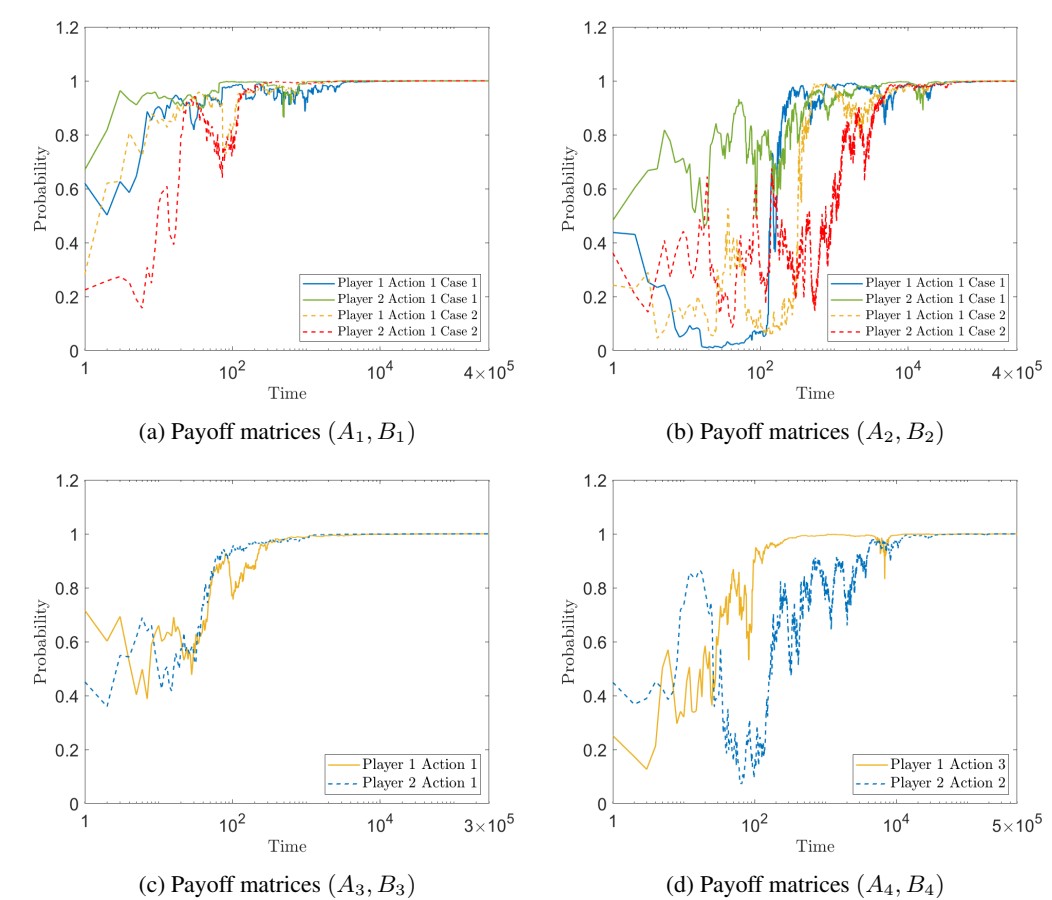

Figure 1: The probability that two players choose the specified action in different game settings.

Moreover, we also report that it can be generalized to the misspecified case, i.e., when the actual noise distribution is not consistent with the Bayesian updating rule in Thompson sampling. For example, Thompson sampling may assume Gaussian noise in the algorithm but the actual noise can be Bernoulli random variables. The payoff matrices are $A_3 = \begin{pmatrix} 0.5 & 0.4 \\ 0.2 & 0.3 \end{pmatrix}$, $B_3 = \begin{pmatrix} 0.7 & 0.3 \\ 0.6 & 0.5 \end{pmatrix}$ with the unique pure-strategy Nash equilibrium $(1, 1)$. We show that the game still converges to the NE. See Figure 1c. We have also relaxed the assumption on two actions and shown the convergence holds under the multiple actions setting in Figure 1d with $A_4 = \begin{pmatrix} 0.6 & 0.4 & 0.1 \\ 0.2 & 0.5 & 0.3 \\ 0.8 & 0.7 & 0.4 \end{pmatrix}$, $B_4 = \begin{pmatrix} 0.4 & 0.6 & 0 \\ 0.3 & 0.3 & 0.6 \\ 0.5 & 0.6 & 0.4 \end{pmatrix}$, whose unique pure Nash equilibrium is $(3, 2)$.

# 7 CONCLUSION AND FUTURE WORK

In this paper, we study a two-player blindfolded game, where both players use Thompson sampling to choose between two actions.. We show that the the game dynamics converge to the pure-strategy Nash equilibrium under mild conditions and algorithmic collusion does not arise.

This study, for the purpose of exposition and clean analysis, makes a number of simplifying assumptions, including limiting our scope to normal conjugate priors, two players and two actions. We hope to extend the analysis to a general setting with general distributions, multiple players and actions in the future research. It is also an open question whether Thompson sampling can converge to the Nash equilibrium in the absence of Assumption 2.

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

# A APPENDIX

## A.1 ADDITIONAL EXPERIMENTS

We conduct additional experiments to illustrate the behavior of the game dynamics when there is no unique pure-strategy Nash Equilirium (NE). The setups of the new numerical experiments are given below.

- Two pure-strategy NE and one mixed-strategy NE. In this experiment, we consider the following payoff matrices A = [0.3 0.3; 0.4 0.1], B = [0.1 0.3; 0.4 0.3]. The game has two pure-strategy NE (action 1, action 2), (action 2, action 1) and one mixed-strategy NE (1/3, 2/3). We simulate 100 sample paths, and find that the game may converge to one of pure-strategy Nash equilibriums:it converges to (action 1, action 2) with probability 78% and (action 2, action 1) with 22%.

- No pure-strategy NE and one mixed-strategy NE. We use the payoff matrices A = [0.5 0.2; 0.1 0.3], B = [0.3 0.5; 0.7 0.4]. From Figure 2, we can see that although the posterior means converge, the probability of action 1 of both players may oscillate. This is because it converges to a point in the probability space that is not Lipschitz continuous. It is unclear whether the empirical distribution of the actions converge to the mixed-strategy NE.

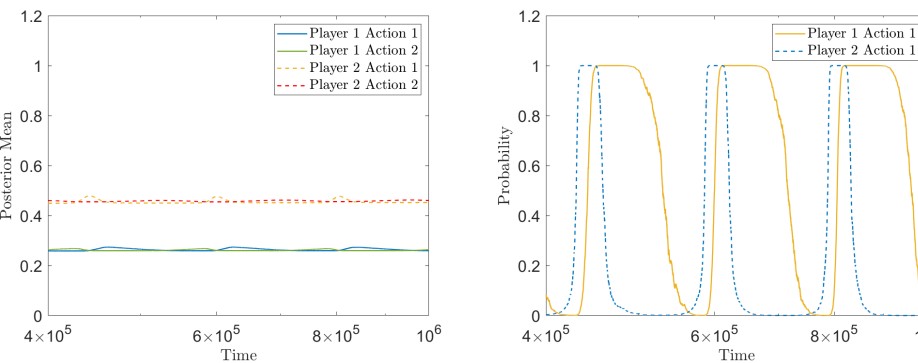

(a) Posterior mean of each action converges.  (b) Probability of each action oscillates.

Figure 2: Game with no pure-strategy NE and one mixed-strategy NE.

## A.2 PROOF OF LEMMA 1

*Proof of Lemma 1.* Without loss of generality, we show $\lim_{n\to\infty} N_{1,n} = \infty$ almost surely. The arguments for proving $\lim_{n\to\infty} N_{2,n} = \infty$ and $\lim_{n\to\infty} M_{j,n} = \infty$ almost surely for $j = 1, 2$ are similar.

Let $E_{1,n}$ denote the event that action 1 is played by player one in round $n$. Then we have

$$\{ \lim_{n\to\infty} N_{1,n} = \infty \} = \{E_{1,n} \ i.o.\}, \tag{21}$$

where $i.o.$ stands for infinitely often. It is clear that $E_{1,n} \in \mathcal{F}_n$, where $\mathcal{F}_n$ is the information set up to time $n$. From the second Borel-Cantelli Lemma (Theorem 5.3.2 in Durrett (2019)), we know that

$$\{E_{1,n} \ i.o.\} = \left\{ \sum_{n=1}^{\infty} \mathsf{P}(E_{1,n}|\mathcal{F}_{n-1}) = \infty \right\} = \left\{ \sum_{n=1}^{+\infty} \varphi_{1,n} = \infty \right\}, \tag{22}$$

where the second equality holds because $\mathsf{P}(E_{1,n}|\mathcal{F}_{n-1})$ is exactly $\varphi_{1,n}$, the probability that action 1 will be chosen by player one at time $n$.

Our goal is to show that $\mathsf{P}(\lim_{n\to\infty} N_{1,n} = \infty) = 1$, or equivalently, $\mathsf{P}(\lim_{n\to\infty} N_{1,n} < \infty) = 0$. Consider any sample path $\boldsymbol{\omega} \in \{\lim_{n\to\infty} N_{1,n} < \infty\}$, and denote by $\overline{N}_1(\boldsymbol{\omega}) :=$

$\lim_{n\to\infty} N_{1,n}(\boldsymbol{\omega}) < \infty$. We analyze below the sum $\sum_{n=1}^{+\infty} \varphi_{1,n}$ on such a path $\boldsymbol{\omega}$, and show that such a path is in a probability zero set.

From Equation (5), we know that

$$\varphi_{1,n} = \mathsf{P}(i_n = 1|\mathcal{F}_{n-1}) = 1 - \Phi\left(\frac{x_{2,n-1} - x_{1,n-1}}{\sqrt{w_{1,n-1} + w_{2,n-1}}}\right),$$

where $\varphi_{1,n}$ depends on the path $\boldsymbol{\omega}$. If $x_{2,n-1} < x_{1,n-1}$ on such a path, then we have $\varphi_{1,n} \geq 1/2$. On the other hand, if $x_{2,n-1} \geq x_{1,n-1}$, then we can use the following tail probability estimate for normal distributions (Formula 7.1.13 in Abramowitz & Stegun (1948)) to bound $\varphi_{1,n}$:

$$\sqrt{\frac{2}{\pi}}e^{-x^2/2}\frac{1}{x + \sqrt{x^2 + 4}} \leq 1 - \Phi(x) \leq \sqrt{\frac{2}{\pi}}e^{-x^2/2}\frac{1}{x + \sqrt{x^2 + 8/\pi}}, \quad x \geq 0.$$

Specifically, we have

$$\varphi_{1,n} \geq \sqrt{\frac{2}{\pi}}e^{-C_n^2/2}\frac{1}{C_n + \sqrt{C_n^2 + 4}}, \tag{23}$$

where $C_n := \frac{x_{2,n-1} - x_{1,n-1}}{\sqrt{w_{1,n-1} + w_{2,n-1}}} \geq 0$.

We next upper bound $C_n$ on the path $\boldsymbol{\omega}$ with $x_{2,n-1} \geq x_{1,n-1}$ to obtain a more tractable bound (uniform in $n$) for $\varphi_{1,n}$. Note that $a_{i_s,s}$ is the random reward, which follows a normal distribution $\mathcal{N}(A_{i_s,j_s}, 1)$ given $i_s$ and $j_s$. We can write $a_{i_s,s} = A_{i_s,j_s} + \xi_s$, where $(\xi_s : s \geq 1)$ is a sequence of i.i.d. standard normal random variables. Let $\bar{A} = \max_{i,j} A_{i,j}$ and $\underline{A} = \min_{i,j} A_{i,j}$. Then we can infer from (1) that

$$x_{2,n-1} = \frac{\sum_{s=1}^{n-1} a_{i_s,s} \cdot \mathbf{1}_{\{i_s=2\}}}{N_{2,n-1} + 1} \leq \bar{A} + \frac{\sum_{s=1}^{n-1} |\xi_s| \cdot \mathbf{1}_{\{i_s=2\}}}{N_{2,n-1} + 1},$$

$$x_{1,n-1} = \frac{\sum_{s=1}^{n-1} a_{i_s,s} \cdot \mathbf{1}_{\{i_s=1\}}}{N_{1,n-1} + 1} \geq \underline{A} - \frac{\sum_{s=1}^{n-1} |\xi_s| \cdot \mathbf{1}_{\{i_s=1\}}}{N_{1,n-1} + 1}.$$

It follows that

$$x_{2,n-1} - x_{1,n-1} \leq \bar{A} - \underline{A} + \frac{\sum_{s=1}^{n-1} |\xi_s| \cdot \mathbf{1}_{\{i_s=2\}}}{N_{2,n-1} + 1} + \frac{\sum_{s=1}^{n-1} |\xi_s| \cdot \mathbf{1}_{\{i_s=1\}}}{N_{1,n-1} + 1}.$$

Recall that we assume on the path $\boldsymbol{\omega}$ we have $\overline{N}_1(\boldsymbol{\omega}) := \lim_{n\to\infty} N_{1,n}(\boldsymbol{\omega}) < \infty$. Because player one can choose only two actions, this implies that $\lim_{n\to\infty} N_{2,n}(\boldsymbol{\omega}) = \infty$. By the strong law of large numbers we obtain that for path $\boldsymbol{\omega}$, $\lim_{n\to\infty} \frac{\sum_{s=1}^{n} |\xi_s| \cdot \mathbf{1}_{\{i_s=2\}}}{N_{2,n}+1} = \mathbb{E}[|\xi_1|] < \infty$, which implies that on the path $\boldsymbol{\omega}$ the sequence $\{\frac{\sum_{s=1}^{n} |\xi_s| \cdot \mathbf{1}_{\{i_s=2\}}}{N_{2,n}+1} : n \geq 1\}$ is bounded. In addition, the sequence $\{\frac{\sum_{s=1}^{n} |\xi_s| \cdot \mathbf{1}_{\{i_s=1\}}}{N_{1,n}+1} : n \geq 1\}$ is also bounded on the path $\boldsymbol{\omega}$ because $\overline{N}_1(\boldsymbol{\omega}) := \lim_{n\to\infty} N_{1,n}(\boldsymbol{\omega}) < \infty$, which implies that there are only finite number of different terms in this sequence. Therefore we can infer that there exists some positive constant $C$ (which depends on $\boldsymbol{\omega}$ but is independent of $n$) such that

$$x_{2,n-1} - x_{1,n-1} \leq \bar{A} - \underline{A} + C.$$

In addition, using the fact that $\overline{N}_1(\boldsymbol{\omega}) := \lim_{n\to\infty} N_{1,n}(\boldsymbol{\omega}) < \infty$, we can obtain on the path $\boldsymbol{\omega}$, $w_{1,n-1} = \frac{1}{N_{1,n-1}+1} \geq \frac{1}{\overline{N}_1(\boldsymbol{\omega})+1} > 0$, which further implies that $w_{1,n-1} + w_{2,n-1} \geq w_{1,n-1} \geq \frac{1}{\overline{N}_1(\boldsymbol{\omega})+1} > 0$. Therefore we can upper bound $C_n$ by

$$C_n := \frac{x_{2,n-1} - x_{1,n-1}}{\sqrt{w_{1,n-1} + w_{2,n-1}}} \leq [\bar{A} - \underline{A} + C] \cdot \sqrt{\overline{N}_1(\boldsymbol{\omega}) + 1} := \bar{C} < \infty.$$

By (23) we then infer that when $x_{2,n-1} - x_{1,n-1} \geq 0$,

$$\varphi_{1,n} \geq \sqrt{\frac{2}{\pi}}e^{-\bar{C}^2/2}\frac{1}{\bar{C} + \sqrt{\bar{C}^2 + 4}} > 0.$$

At any time $n-1$, we know that either $x_{2,n-1} < x_{1,n-1}$ or $x_{2,n-1} \geq x_{1,n-1}$ holds for the path $\boldsymbol{\omega}$. Therefore, if let $\bar{\varphi} := \min\left\{\frac{1}{2}, \sqrt{\frac{2}{\pi}} e^{-\bar{C}^2/2} \frac{1}{\bar{C}+\sqrt{\bar{C}^2+4}}\right\} > 0$, we then have $\varphi_{1,n} \geq \bar{\varphi}$ for all $n$ on the path $\boldsymbol{\omega}$. As a consequence, we obtain that for any sample path $\boldsymbol{\omega} \in \{\lim_{n\to\infty} N_{1,n} < \infty\}$,

$$\sum_{n=1}^{\infty} \varphi_{1,n} \geq \sum_{n=1}^{\infty} \bar{\varphi} = \infty.$$

This suggests that

$$\left\{\lim_{n\to\infty} N_{1,n} < \infty\right\} \subset \left\{\sum_{n=1}^{+\infty} \varphi_{1,n} = \infty\right\}.$$

However, from (21) and (22) we have

$$\left\{\sum_{n=1}^{+\infty} \varphi_{1,n} = \infty\right\} = \{\lim_{n\to\infty} N_{1,n} = \infty\}.$$

Thus the set $\{\lim_{n\to\infty} N_{1,n} < \infty\}$ has probability zero. We then conclude that the event $\{\lim_{n\to\infty} N_{1,n} = \infty\}$ holds with probability one. The proof is therefore complete. $\qquad\square$

### A.3 Proof of Lemma 2

*Proof of Lemma 2.* We show that for $i = 1, 2$, $\sum_{n=1}^{+\infty} \alpha_{i,n} = \infty$ and $\sum_{n=1}^{+\infty} \alpha_{i,n}^2 < \infty$ almost surely. The arguments for $(\beta_{j,n})_{j=1,2}$ are completely analogous. Recall from Section 2.2 that $\alpha_{i,n}$ are binary-valued random variables with $\alpha_{i,n} = \frac{1}{N_{i,n}+1}$ if action $i$ is selected by player one in round $n$ and $\alpha_{i,n} = 0$ otherwise. Fix any one sample path in the probability one set where Lemma 1 holds. Suppose at time $s$, action $i$ is chosen and $N_{i,s-1} = a$ for some $a \in \mathbb{N}$, then we have $N_{i,s} = a + 1$ and $\alpha_{i,s} = \frac{1}{N_{i,s}+1} = \frac{1}{a+2}$ on this path. Lemma 1 shows that action $i$ is chosen infinitely often, so there exists some time $\tau > s$ that action $i$ is chosen again, and $\alpha_{i,\tau} = \frac{1}{N_{i,\tau}+1} = \frac{1}{(a+2)+1} = \frac{1}{a+3}$. Repeating this argument, we can infer that $\{\frac{1}{n}\}_{n=1}^{+\infty}$ is a subsequence of $\{\alpha_{i,n}\}_{n=1}^{\infty}$ on such a sample path, where other elements in the sequence $\{\alpha_{i,n}\}_{n=1}^{\infty}$ are all zero. Therefore, we obtain $\sum_{n=1}^{+\infty} \alpha_{i,n} = \sum_{n=1}^{+\infty} 1/n = \infty$ and $\sum_{n=1}^{+\infty} \alpha_{i,n}^2 = \sum_{n=1}^{+\infty} 1/n^2 < \infty$ on such a sample path. This completes the proof of Lemma 2. $\qquad\square$

### A.4 Proof of Lemma 3

*Proof of Lemma 3.* It is straightforward to obtain from (8) and (7) that for $i = 1, 2$, $\mathbb{E}[\bar{a}_{i,n+1}|\mathcal{F}_n] = 0$ for all $n$. Similarly, we can infer from (12) and (5) that $\mathbb{E}[\bar{b}_{j,n+1}|\mathcal{F}_n] = 0$ for all $n$ and $j = 1, 2$. Hence we obtain that $\mathbb{E}[\bar{\boldsymbol{\xi}}_{n+1}|\mathcal{F}_n] = 0$ for all $n$.

In addition, we can directly compute from (8) that

$$\mathbb{E}[\bar{a}_{i,n+1}^2|\mathcal{F}_n] = 1 + \mathbb{E}\left[(\sum_{j=1}^{2} A_{i,j}\mathbf{1}_{\{j_{n+1}=j\}} - \sum_{j=1}^{2} A_{i,j}\psi_{j,n+1})^2 \Big| \mathcal{F}_n\right]$$
$$= 1 + [A_{i,1} - A_{i,2}]^2 \cdot \psi_{1,n+1}(1 - \psi_{1,n+1})$$
$$\leq 1 + [A_{i,1} - A_{i,2}]^2/4,$$

where the second equality follows from the fact that given $\mathcal{F}_n$, $\mathbf{1}_{\{j_{n+1}=j\}}$ is a Bernoulli random variable. Similarly, we have

$$\mathbb{E}[\bar{b}_{j,n+1}^2|\mathcal{F}_n] = 1 + \mathbb{E}\left[(\sum_{i=1}^{2} B_{i,j}\mathbf{1}_{\{i_{n+1}=i\}} - \sum_{i=1}^{2} B_{i,j}\varphi_{i,n+1})^2 \Big| \mathcal{F}_n\right]$$
$$\leq 1 + [B_{1,j} - B_{2,j}]^2/4.$$

Therefore we obtain from (16) that for all $n$,

$$E[\bar{\xi}_{n+1}^2|\mathcal{F}_n] = \sum_i \mathbb{E}[\bar{a}_{i,n+1}^2|\mathcal{F}_n] + \sum_j \mathbb{E}[\bar{b}_{j,n+1}^2|\mathcal{F}_n] \le 4 + [A_{i,1} - A_{i,2}]^2/2 + [B_{1,j} - B_{2,j}]^2/2.$$

The proof is then complete. $\qquad\square$

## A.5 Proof of Lemma 4

*Proof of Lemma 4.* Recall from (3) that $\boldsymbol{S}_n := (x_{1,n}, x_{2,n}, y_{1,n}, y_{2,n}, w_{1,n}, w_{2,n}, v_{1,n}, v_{2,n}) \in \mathbb{R}^4 \times \mathbb{R}_+^4$. From the definitions in (1) and (2), we obtain that and $|w_{i,n}|$ and $|v_{j,n}|$ are bounded by 1 for $i, j \in \{1, 2\}$. Then we have

$$\sup_n \|\boldsymbol{S}_n\| \le \sum_{i=1}^2 \sup_n |x_{i,n}| + \sum_{j=1}^2 \sup_n |y_{j,n}| + 4.$$

We first prove that $\sup_n |x_{i,n}| < \infty$ almost surely for $i = 1, 2$. Recall from (1) that

$$x_{i,n} := \frac{\sum_{s=1}^n a_{i_s,s} \cdot \mathbf{1}_{\{i_s=i\}}}{N_{i,n} + 1},$$

where $a_{i_s,s}$ is the random reward following a normal distribution $\mathcal{N}(A_{i_s,j_s}, 1)$, and $N_{i,n} = \sum_{s=1}^n \mathbf{1}_{\{i_s=i\}}$ denotes the number of plays of action $i$ by Player 1 up to round $n$. We can write $a_{i_s,s} = A_{i_s,j_s} + \xi_s$, where $(\xi_s : s \ge 1)$ is a sequence of i.i.d. standard normal random variables. Hence, we have

$$|x_{i,n}| \le \frac{\sum_{s=1}^n |a_{i_s,s}| \cdot \mathbf{1}_{\{i_s=i\}}}{N_{i,n} + 1} \le \max_{i,j \in \{1,2\}} |A_{i,j}| + \frac{\sum_{s=1}^n |\xi_s| \cdot \mathbf{1}_{\{i_s=i\}}}{N_{i,n} + 1}.$$

It follows that

$$\sup_n |x_{i,n}| \le \max_{i,j \in \{1,2\}} |A_{i,j}| + \sup_n \frac{\sum_{s=1}^n |\xi_s| \cdot \mathbf{1}_{\{i_s=i\}}}{N_{i,n} + 1}. \qquad (24)$$

For each sample path, $\{N_{i,n} : n \ge 1\}$ is a non-decreasing sequence of integers and hence we can set $N_{i,\infty} = \lim_{n \to \infty} N_{i,n}$. In view of (24), to show $\sup_n |x_{i,n}| < \infty$ almost surely, it suffices to consider those sample paths with $N_{i,\infty} = \infty$. For each of such sample paths (except a possible zero-probability set), we can infer from the strong law of large numbers that $\lim_{n\to\infty} \frac{\sum_{s=1}^n |\xi_s| \cdot \mathbf{1}_{\{i_s=i\}}}{N_{i,n}+1} = \mathbb{E}[|\xi_1|] < \infty$. This implies that $\sup_n \frac{\sum_{s=1}^n |\xi_s| \cdot \mathbf{1}_{\{i_s=i\}}}{N_{i,n}+1} < \infty$ on such paths. Therefore, we can infer from (24) that $\sup_n |x_{i,n}| < \infty$ almost surely.

Similarly, we can prove that $\sup_n |y_{j,n}| < \infty$ almost surely for $j = 1, 2$. Thus, we obtain $\sup_n \|\boldsymbol{S}_n\| < \infty$ almost surely. The proof is hence complete. $\qquad\square$

## A.6 Proof of Theorem 1

*Proof of Theorem 1.* The proof is based on a sample-path-wise argument. For $n \ge 1$, we first rewrite the dynamics (15) to the following recursion form

$$\boldsymbol{S}_n - \boldsymbol{S}^* = (1 - \boldsymbol{\gamma}_n) \circ (\boldsymbol{S}_{n-1} - \boldsymbol{S}^*) + \boldsymbol{\gamma}_n \circ \left(F(\boldsymbol{S}_{n-1}) - \boldsymbol{S}^* + \bar{\boldsymbol{\xi}}_n\right), \qquad (25)$$

where $\boldsymbol{S}^*$ is defined in (17). Denote $S_{k,n}$ as the k-th entry of $\boldsymbol{S}_n$ ($\gamma_{k,n}$ is the k-th entry of $\boldsymbol{\gamma}_n$). We state three preliminary lemmas, the proofs of which are given in Appendix A.7, A.8 and A.9.

**Lemma 5.** *For $k = 1, \ldots, 8$, let $\prod_{s=n+1}^n (1 - \gamma_{k,s}) = 1$ by convention, then for any $0 \le m \le n$, $S_{k,n} - S_k^*$ has the following recursive form almost surely*

$$S_{k,n} - S_k^* = (S_{k,m} - S_k^*) \cdot \prod_{\tau=m+1}^n (1 - \gamma_{k,\tau}) + \sum_{\tau=m+1}^n \left[\prod_{s=\tau+1}^n (1 - \gamma_{k,s})\right] \gamma_{k,\tau} \left(F_k(\boldsymbol{S}_{\tau-1}) - S_k^* + \bar{\xi}_{k,\tau}\right).$$

**Lemma 6.** *For $k = 1, \ldots, 8$, let $\prod_{s=n+1}^{n}(1 - \gamma_{k,s}) = 1$ by convention, then for any $1 \leq m \leq n$, we have*

$$\prod_{\tau=m}^{n}(1 - \gamma_{k,\tau}) + \sum_{\tau=m}^{n}\left[\prod_{s=\tau+1}^{n}(1 - \gamma_{k,s})\right]\gamma_{k,\tau} = 1, \text{ almost surely.}$$

**Lemma 7.** *For $k = 1, \ldots, 8$, we have $\prod_{\tau=1}^{\infty}(1 - \gamma_{k,\tau}) = 0$ almost surely.*

Now we present the proof of Theorem 1, which builds on the proof of Theorem 3 in (Tsitsiklis, 1994). Fix any sample path $\omega$ (that does not lie in the null sets in the three lemmas above) throughout the proof. For notational simplicity, we omit the specification of the path $\omega$ below.

We first show that $\boldsymbol{S}_n$ will avoid the region where $F$ is not Lipschitz continuous for a sufficiently large $n$. Consider the recursion of $\boldsymbol{S}_n(n \geq 1)$ starting from period 0, from Lemma 5, for any entry $k$, we have

$$S_{k,n} - S_k^* = (S_{k,0} - S_k^*) \cdot \prod_{\tau=1}^{n}(1 - \gamma_{k,\tau}) + \sum_{\tau=1}^{n}\left[\prod_{s=\tau+1}^{n}(1 - \gamma_{k,s})\right]\gamma_{k,\tau}\left(F_k(\boldsymbol{S}_{\tau-1}) - S_k^* + \bar{\xi}_{k,\tau}\right).$$

$$(26)$$

For $n \geq 1$, let

$$C_{k,n} := (S_{k,0} - S_k^*) \cdot \prod_{\tau=1}^{n}(1 - \gamma_{k,\tau}),$$

$$D_{k,n} := \sum_{\tau=1}^{n}\left[\prod_{s=\tau+1}^{n}(1 - \gamma_{k,s})\right]\gamma_{k,\tau}(F_k(\boldsymbol{S}_{\tau-1}) - S_k^*),$$

$$E_{k,n} := \sum_{\tau=1}^{n}\left[\prod_{s=\tau+1}^{n}(1 - \gamma_{k,s})\right]\gamma_{k,\tau}\bar{\xi}_{k,\tau},$$

Then (26) implies that

$$S_{k,n} - S_k^* = C_{k,n} + D_{k,n} + E_{k,n}, \quad k = 1, \ldots, 8. \tag{27}$$

For the first term $C_{k,n}$, we can apply Lemma 7, and obtain

$$\lim_{n\to\infty} C_{k,n} = 0, \quad \forall k. \tag{28}$$

Next, let us consider the third term $E_{k,n}$. Recall the definition of $\bar{\xi}_n$ in (16), we know $\bar{\xi}_{k,n} = 0$ for any $n \geq 1$, $k = 5, 6, 7, 8$, which implies $E_{k,n} = 0$ for $k = 5, 6, 7, 8$. Moreover, for any $1 \leq m \leq n - 1$, $E_{k,n}$ has the following recursion for $0 \leq m \leq n$

$$E_{k,n} = \prod_{\tau=m+1}^{n}(1 - \gamma_{k,\tau}) \cdot E_{k,m} + \sum_{\tau=m+1}^{n}\left[\prod_{s=\tau+1}^{n}(1 - \gamma_{k,s})\right]\gamma_{k,\tau}\bar{\xi}_{k,\tau}.$$

From the proof of Lemma 2 in Tsitsiklis (1994), we immediately have

$$\lim_{n\to\infty} E_{k,n} = 0, \ k = 1, 2, 3, 4. \tag{29}$$

Finally, we discuss the remaining second term $D_{k,n}$. For $k = 5, 6, 7, 8$, by the definition of $F(\boldsymbol{S})$ in (13), we know $F_k(\boldsymbol{S}_n) = 0$ for any $n \geq 1$. Moreover, we know $S_k^* = 0$ from (17), so $D_{k,n} = 0$ for $k = 5, 6, 7, 8$. Therefore, we only need to consider $D_{k,n}$ for $k = 1, 2, 3, 4$.

Note that

$$D_{1,n} = \sum_{\tau=1}^{n}\left[\prod_{s=\tau+1}^{n}(1 - \gamma_{1,s})\right]\gamma_{1,\tau}(F_1(\boldsymbol{S}_{\tau-1}) - S_1^*)$$

$$= \sum_{\tau=1}^{n}\left[\prod_{s=\tau+1}^{n}(1 - \gamma_{1,s})\right]\gamma_{1,\tau}(A_{1,1}\psi_{1,\tau} + A_{1,2}\psi_{2,\tau} - A_{1,1})$$

$$= \sum_{\tau=1}^{n}\left[\prod_{s=\tau+1}^{n}(1 - \gamma_{1,s})\right]\gamma_{1,\tau}(A_{1,2} - A_{1,1})\psi_{2,\tau}.$$

It follows that

$$|D_{1,n}| \le |A_{1,2} - A_{1,1}| \cdot \sum_{\tau=1}^{n} \prod_{s=\tau+1}^{n} (1 - \gamma_{1,s}) \gamma_{1,\tau}$$

$$= |A_{1,2} - A_{1,1}| \cdot \left[ 1 - \prod_{\tau=1}^{n} (1 - \gamma_{k,\tau}) \right]$$

$$\le |A_{1,2} - A_{1,1}|, \tag{30}$$

where the equation holds due to Lemma 6. Similarly, we have

$$|D_{2,n}| = \left| \sum_{\tau=1}^{n} \left[ \prod_{s=\tau+1}^{n} (1 - \gamma_{2,s}) \right] \gamma_{2,\tau} (A_{2,2} - A_{2,1}) \psi_{2,\tau} \right| \le |A_{2,2} - A_{2,1}|,$$

$$|D_{3,n}| = \left| \sum_{\tau=1}^{n} \left[ \prod_{s=\tau+1}^{n} (1 - \gamma_{3,s}) \right] \gamma_{3,\tau} (B_{2,1} - B_{1,1}) \varphi_{2,\tau} \right| \le |B_{2,1} - B_{1,1}|,$$

$$|D_{4,n}| = \left| \sum_{\tau=1}^{n} \left[ \prod_{s=\tau+1}^{n} (1 - \gamma_{4,s}) \right] \gamma_{4,\tau} (B_{2,2} - B_{1,2}) \varphi_{2,\tau} \right| \le |B_{2,2} - B_{1,2}|. \tag{31}$$

From Assumption 2, we can obtain there exists $\epsilon_1 \in (0,1)$ and $\epsilon_2 \in (0,1)$ such that

$$|A_{1,2} - A_{1,1}| + |A_{2,2} - A_{2,1}| \le (1 - \epsilon_1)(A_{1,1} - A_{2,1}),$$
$$|B_{2,1} - B_{1,1}| + |B_{2,2} - B_{1,2}| \le (1 - \epsilon_2)(B_{1,1} - B_{1,2}). \tag{32}$$

Recall $\lim_{n \to \infty} C_{k,n} = 0$ for all $k$ in (28). Given $0 < \epsilon_3 < \frac{\min\{\epsilon_1, \epsilon_2\}}{2}$, we can obtain that there exists $n_0$ such that for $n > n_0$,

$$|C_{k,n}| \le \frac{\epsilon_3}{4}(A_{1,1} - A_{2,1}), \text{ for } k = 1, 2.$$

$$|C_{k,n}| \le \frac{\epsilon_3}{4}(B_{1,1} - B_{1,2}), \text{ for } k = 3, 4.$$

$$|C_{k,n}| \le \frac{\epsilon_3}{4}(A_{1,1} - A_{2,1}), \text{ for } k = 5, 6.$$

$$|C_{k,n}| \le \frac{\epsilon_3}{4}(B_{1,1} - B_{1,2}), \text{ for } k = 7, 8. \tag{33}$$

By (29), we have $\lim_{n \to \infty} E_{k,n} = 0$ for $k = 1, 2, 3, 4$. Therefore, there exists $\tau_0$ such that for $n > \tau_0$, there holds

$$|E_{k,n}| \le \frac{\epsilon_3}{4}(A_{1,1} - A_{2,1}), \text{ for } k = 1, 2.$$

$$|E_{k,n}| \le \frac{\epsilon_3}{4}(B_{1,1} - B_{1,2}), \text{ for } k = 3, 4. \tag{34}$$

Therefore, for $n > \max\{n_0, \tau_0\}$. we can infer from (27), (33) and (34) that

$$|x_{1,n} - A_{1,1}| \le |C_{1,n}| + |D_{1,n}| + |E_{1,n}|$$

$$\le \frac{\epsilon_3}{4}(A_{1,1} - A_{2,1}) + |A_{1,2} - A_{1,1}| + \frac{\epsilon_3}{4}(A_{1,1} - A_{2,1})$$

$$= \frac{\epsilon_3}{2}(A_{1,1} - A_{2,1}) + |A_{1,2} - A_{1,1}|.$$

Similarly,

$$|x_{2,n} - A_{2,1}| \le \frac{\epsilon_3}{2}(A_{1,1} - A_{2,1}) + |A_{2,2} - A_{2,1}|,$$

$$|y_{1,n} - B_{1,1}| \le \frac{\epsilon_3}{2}(B_{1,1} - B_{1,2}) + |B_{2,1} - B_{1,1}|,$$

$$|y_{2,n} - B_{1,2}| \le \frac{\epsilon_3}{2}(B_{1,1} - B_{1,2}) + |B_{2,2} - B_{1,2}|.$$

Therefore, for $n \geq \max\{n_0, \tau_0\}$, we have

$$
\begin{aligned}
&x_{1,n} - x_{2,n} \\
&\geq A_{1,1} - \left[\frac{\epsilon_3}{2}(A_{1,1} - A_{2,1}) + |A_{1,2} - A_{1,1}|\right] - \left[A_{2,1} + \frac{\epsilon_3}{2}(A_{1,1} - A_{2,1}) + |A_{2,2} - A_{2,1}|\right] \\
&= (1 - \epsilon_3)(A_{1,1} - A_{2,1}) - (|A_{1,2} - A_{1,1}| + |A_{2,2} - A_{2,1}|),
\end{aligned}
\tag{35}
$$

and

$$
\begin{aligned}
&y_{1,n} - y_{2,n} \\
&\geq B_{1,1} - \left[\frac{\epsilon_3}{2}(B_{1,1} - B_{1,2}) + |B_{2,1} - B_{1,1}|\right] - \left[B_{1,2} + \frac{\epsilon_3}{2}(B_{1,1} - B_{1,2}) + |B_{2,2} - B_{1,2}|\right] \\
&= (1 - \epsilon_3)(B_{1,1} - B_{1,2}) - (|B_{2,1} - B_{1,1}| + |B_{2,2} - B_{1,2}|)
\end{aligned}
\tag{36}
$$

Note that $0 < \epsilon_3 < \frac{\min\{\epsilon_1, \epsilon_2\}}{2}$, from Assumption 2 and formulas (32), for $n \geq \max\{n_0, \tau_0\}$, (35) and (36) can be lower bounded by

$$
x_{1,n} - x_{2,n} \geq (\epsilon_1 - \epsilon_3)(A_{1,1} - A_{2,1}) > \frac{\epsilon_1}{2}(A_{1,1} - A_{2,1}) > 0,
$$

$$
y_{1,n} - y_{2,n} \geq (\epsilon_2 - \epsilon_3)(B_{1,1} - B_{1,2}) > \frac{\epsilon_2}{2}(B_{1,1} - B_{1,2}) > 0.
$$

Recall $\varphi_{1,n+1} = \Phi\left(\frac{x_{1,n} - x_{2,n}}{\sqrt{w_{1,n} + w_{2,n}}}\right)$ and $\psi_{1,n+1} = \Phi\left(\frac{y_{1,n} - y_{2,n}}{\sqrt{v_{1,n} + v_{2,n}}}\right)$. What we have shown is that $\boldsymbol{S}_n$ will avoid the region where $F$ is not Lipschitz continuous for a sufficiently large $n$.

Next, to apply the convergence result (Theorem 3) in (Tsitsiklis, 1994), we need to guarantee the Lipschitz constant of $F$ is smaller than 1, *i.e.*, to prove there exist $\delta \in [0, 1)$ such that for $n$ large enough,

$$
\|F(\boldsymbol{S}_n) - F(\boldsymbol{S}^*)\|_\infty \leq \delta \|\boldsymbol{S}_n - \boldsymbol{S}^*\|_\infty.
$$

To prove the results, we apply the mean value theorem. We have for $i = 1, \ldots, 4$,

$$
F_i(\boldsymbol{S}_n) - F_i(\boldsymbol{S}^*) = \nabla F_i(\tilde{\boldsymbol{S}}_n) \cdot (\boldsymbol{S}_n - \boldsymbol{S}^*),
\tag{37}
$$

where $\tilde{\boldsymbol{S}}_n = (\tilde{x}_{1,n}, \tilde{x}_{2,n}, \tilde{y}_{1,n}, \tilde{y}_{2,n}, \tilde{w}_{1,n}, \tilde{w}_{2,n}, \tilde{v}_{1,n}, \tilde{v}_{2,n})$ is a point on the line segment between $\boldsymbol{S}_n$ and $\boldsymbol{S}^*$. Hence it suffices to bound the gradient $\nabla F_i(\tilde{\boldsymbol{S}}_n)$. Write the Jacobian matrix

$$
L = \begin{pmatrix} L_{1,1}^n & L_{1,2}^n & \cdots & L_{1,8}^n \\ L_{2,1}^n & L_{2,2}^n & \cdots & L_{2,8}^n \\ L_{3,1}^n & L_{3,2}^n & \cdots & L_{3,8}^n \\ L_{4,1}^n & L_{4,2}^n & \cdots & L_{4,8}^n \end{pmatrix} \in \mathbb{R}^{4 \times 8},
$$

where $L_{i,j} = \frac{\partial F_i(\boldsymbol{S})}{\partial S_j}|_{\boldsymbol{S} = \tilde{\boldsymbol{S}}}, i = 1, \ldots, 4, j = 1, \ldots, 8$. By the definition of $F$ in (13), it is easy to see that

$$
\begin{aligned}
L_{1,1}^n = L_{1,2}^n = L_{1,5}^n = L_{1,6}^n = 0, \\
L_{2,1}^n = L_{2,2}^n = L_{2,5}^n = L_{2,6}^n = 0, \\
L_{3,3}^n = L_{3,4}^n = L_{3,7}^n = L_{3,8}^n = 0, \\
L_{4,3}^n = L_{4,4}^n = L_{4,7}^n = L_{4,8}^n = 0.
\end{aligned}
$$

To bound other $L_{ij}^n$ terms, we recall the definition of $\boldsymbol{S}^*$ in (17), and note that there exists $\rho \in [0, 1]$, such that $\tilde{\boldsymbol{S}}_n = \rho \boldsymbol{S}_n + (1 - \rho)\boldsymbol{S}^*$. Then for $n > \max\{n_0, \tau_0\}$, we have

$$
\tilde{x}_{1,n} - \tilde{x}_{2,n} = \rho(x_{1,n} - x_{2,n}) + (1 - \rho)(A_{1,1} - A_{2,1}) > \frac{\epsilon_1}{2}(A_{1,1} - A_{2,1}),
$$

$$
\tilde{y}_{1,n} - \tilde{y}_{2,n} = \rho(y_{1,n} - y_{2,n}) + (1 - \rho)(B_{1,1} - B_{1,2}) > \frac{\epsilon_2}{2}(B_{1,1} - B_{1,2}),
$$

$$
\tilde{w}_{1,n} + \tilde{w}_{2,n} = \rho(w_{1,n} + w_{2,n}) + (1 - \rho) \cdot 0 \leq w_{1,n} + w_{2,n},
$$

$$
\tilde{v}_{1,n} + \tilde{v}_{2,n} = \rho(v_{1,n} + v_{2,n}) + (1 - \rho) \cdot 0 \leq v_{1,n} + v_{2,n}.
$$

Let $z_{1,n} := \frac{\tilde{y}_{1,n} - \tilde{y}_{2,n}}{\sqrt{\tilde{v}_{1,n} + \tilde{v}_{2,n}}}$, $z_{2,n} := \frac{\tilde{x}_{1,n} - \tilde{x}_{2,n}}{\sqrt{\tilde{w}_{1,n} + \tilde{w}_{2,n}}}$, then we can directly compute

$$|L_{1,3}^n| = |L_{1,4}^n| = \frac{|A_{1,1} - A_{1,2}|}{\sqrt{2\pi}} \cdot \exp\left[-\frac{1}{2}\left(\frac{\tilde{y}_{1,n} - \tilde{y}_{2,n}}{\sqrt{\tilde{v}_{1,n} + \tilde{v}_{2,n}}}\right)^2\right] \frac{1}{\sqrt{\tilde{v}_{1,n} + \tilde{v}_{2,n}}}$$

$$= \frac{|A_{1,1} - A_{1,2}|}{\sqrt{2\pi}} \cdot e^{-\frac{1}{2}z_{1,n}^2} \cdot z_{1,n} \cdot \frac{1}{\tilde{y}_{1,n} - \tilde{y}_{2,n}}$$

$$\leq \frac{|A_{1,1} - A_{1,2}|}{\sqrt{2\pi}} \cdot e^{-\frac{1}{2}z_{1,n}^2} \cdot z_{1,n} \cdot \frac{2}{\epsilon_2(B_{1,1} - B_{1,2})},$$

where the last inequality is due to $\tilde{y}_{1,n} - \tilde{y}_{2,n} > \frac{\epsilon_2}{2}(B_{1,1} - B_{1,2})$. In addition,

$$|L_{1,7}^n| = |L_{1,8}^n|$$

$$= \frac{|A_{1,1} - A_{1,2}|}{2\sqrt{2\pi}} \cdot \exp\left[-\frac{1}{2}\left(\frac{\tilde{y}_{1,n} - \tilde{y}_{2,n}}{\sqrt{\tilde{v}_{1,n} + \tilde{v}_{2,n}}}\right)^2\right] \frac{\tilde{y}_{1,n} - \tilde{y}_{2,n}}{(\tilde{v}_{1,n} + \tilde{v}_{2,n})^{3/2}}$$

$$= \frac{|A_{1,1} - A_{1,2}|}{2\sqrt{2\pi}} \cdot e^{-\frac{1}{2}z_{1,n}^2} \cdot z_{1,n}^3 \cdot \frac{1}{(\tilde{y}_{1,n} - \tilde{y}_{2,n})^2}$$

$$\leq \frac{|A_{1,1} - A_{1,2}|}{2\sqrt{2\pi}} \cdot e^{-\frac{1}{2}z_{1,n}^2} \cdot z_{1,n}^3 \cdot \frac{4}{\epsilon_2^2(B_{1,1} - B_{1,2})^2}.$$

Similarly,

$$|L_{2,3}^n| = |L_{2,4}^n| \leq \frac{|A_{2,1} - A_{2,2}|}{\sqrt{2\pi}} \cdot e^{-\frac{1}{2}z_{1,n}^2} \cdot z_{1,n} \cdot \frac{2}{\epsilon_2(B_{1,1} - B_{1,2})}.$$

$$|L_{2,7}^n| = |L_{2,8}^n| \leq \frac{|A_{2,1} - A_{2,2}|}{2\sqrt{2\pi}} \cdot e^{-\frac{1}{2}z_{1,n}^2} \cdot z_{1,n}^3 \cdot \frac{4}{\epsilon_2^2(B_{1,1} - B_{1,2})^2}.$$

$$|L_{3,1}^n| = |L_{3,2}^n| \leq \frac{|B_{1,1} - B_{2,1}|}{\sqrt{2\pi}} \cdot e^{-\frac{1}{2}z_{2,n}^2} \cdot z_{2,n} \cdot \frac{2}{\epsilon_1(A_{1,1} - A_{2,1})}.$$

$$|L_{3,5}^n| = |L_{3,6}^n| \leq \frac{|B_{1,1} - B_{2,1}|}{2\sqrt{2\pi}} \cdot e^{-\frac{1}{2}z_{2,n}^2} \cdot z_{2,n}^3 \cdot \frac{4}{\epsilon_1^2(A_{1,1} - A_{2,1})^2}.$$

$$|L_{4,1}^n| = |L_{4,2}^n| \leq \frac{|B_{1,2} - B_{2,2}|}{\sqrt{2\pi}} \cdot e^{-\frac{1}{2}z_{2,n}^2} \cdot z_{2,n} \cdot \frac{2}{\epsilon_1(A_{1,1} - A_{2,1})}.$$

$$|L_{4,5}^n| = |L_{4,6}^n| \leq \frac{|B_{1,2} - B_{2,2}|}{2\sqrt{2\pi}} \cdot e^{-\frac{1}{2}z_{2,n}^2} \cdot z_{2,n}^3 \cdot \frac{4}{\epsilon_1^2(A_{1,1} - A_{2,1})^2}.$$

Let $h_1(z) = e^{-\frac{1}{2}z^2} \cdot z$, which is a decreasing function when $z \in [1, \infty)$ and $z \in (-\infty, -1]$. Let $h_2(z) = e^{-\frac{1}{2}z^2} \cdot z^3$, which is also a decreasing function when $z \in [\sqrt{3}, \infty)$ and $z \in (-\infty, -\sqrt{3}]$. And we have

$$z_{1,n} = \frac{\tilde{y}_{1,n} - \tilde{y}_{2,n}}{\sqrt{\tilde{v}_{1,n} + \tilde{v}_{2,n}}} \geq \frac{\epsilon_2(B_{1,1} - B_{1,2})}{2\sqrt{v_{1,n} + v_{2,n}}} = \frac{\epsilon_2(B_{1,1} - B_{1,2})}{2\sqrt{\frac{1}{M_{1,n}+1} + \frac{1}{M_{2,n}+1}}},$$

$$z_{2,n} = \frac{\tilde{x}_{1,n} - \tilde{x}_{2,n}}{\sqrt{\tilde{w}_{1,n} + \tilde{w}_{2,n}}} \geq \frac{\epsilon_1(A_{1,1} - A_{2,1})}{2\sqrt{w_{1,n} + w_{2,n}}} = \frac{\epsilon_1(A_{1,1} - A_{2,1})}{2\sqrt{\frac{1}{N_{1,n}+1} + \frac{1}{N_{2,n}+1}}}.$$

Denote by $\bar{L}_i^n := \max_{j=1,\ldots,8} L_{i,j}^n$, $i = 1, 2, 3, 4$. From Lemma 1, we have $\lim_{n\to\infty} N_{i,n} = \infty$ ($i = 1, 2$) and $\lim_{n\to\infty} M_{j,n} = \infty$ ($j = 1, 2$) almost surely. So there also exists $\eta_0 > \max\{n_0, \tau_0\}$ such that for $n > \eta_0$,

$$\bar{L}_i^n \leq \frac{1}{16}.$$

By the mean value theorem, from (37), we have

$$|F_i(\boldsymbol{S}_n) - F_i(\boldsymbol{S}^*)| \leq \bar{L}_i^n \sum_{j=1}^{8} |S_{j,n} - S_j^*| \leq 8\bar{L}_i^n \|\boldsymbol{S}_n - \boldsymbol{S}^*\|_\infty \leq \frac{1}{2}\|\boldsymbol{S}_n - \boldsymbol{S}^*\|_\infty.$$

Therefore, we obtain for $n > \eta_0$,

$$\|F(\boldsymbol{S}_n) - F(\boldsymbol{S}^*)\|_\infty \leq \frac{1}{2}\|\boldsymbol{S}_n - \boldsymbol{S}^*\|_\infty.$$

Thus, Assumption 5 in Tsitsiklis (1994) can be satisfied. Then applying Theorem 3 in Tsitsiklis (1994), we can get that $\boldsymbol{S}_n$ converges to $\boldsymbol{S}^*$. $\qquad\square$

A.7    PROOF OF LEMMA 5

*Proof of Lemma 5.* We prove this lemma by induction. Firstly, when $n = m$, this recursion obviously holds. Suppose it holds for time $n = N$, which means that

$$S_{k,N} - S_k^* = (S_{k,m} - S_k^*) \cdot \prod_{\tau=m+1}^{N} (1 - \gamma_{k,\tau}) + \sum_{\tau=m+1}^{N} \left( \prod_{s=\tau+1}^{N} (1 - \gamma_{k,s}) \right) \gamma_{k,\tau} \left( F_k(\boldsymbol{S}_{\tau-1}) - S_k^* + \bar{\xi}_{k,\tau} \right).$$
$$(38)$$

Next consider time $n = N + 1$. From (25), we first have

$$S_{k,N+1} - S_k^* = (1 - \gamma_{k,N+1})(S_{k,N} - S_k^*) + \gamma_{k,N+1} \left( F_k(\boldsymbol{S}_N) - S_k^* + \bar{\xi}_{k,N+1} \right).$$

Based on the assumption for $n = N$, we then replace term $S_{k,N} - S_k^*$ by right hand side of (38):

$$S_{k,N+1} - S_k^*$$
$$= (1 - \gamma_{k,N+1})(S_{k,N} - S_k^*) + \gamma_{k,N+1} \left( F_k(\boldsymbol{S}_N) - S_k^* + \bar{\xi}_{k,N+1} \right)$$
$$= (1 - \gamma_{k,N+1}) \cdot \left[ (S_{k,m} - S_k^*) \cdot \prod_{\tau=m+1}^{N} (1 - \gamma_{k,\tau}) + \sum_{\tau=m+1}^{N} \left( \prod_{s=\tau+1}^{N} (1 - \gamma_{k,s}) \right) \gamma_{k,\tau} \left( F_k(\boldsymbol{S}_{\tau-1}) - S_k^* + \bar{\xi}_{k,\tau} \right) \right]$$
$$+ \gamma_{k,N+1} \left( F_k(\boldsymbol{S}_N) - S_k^* + \bar{\xi}_{k,N+1} \right)$$
$$= (S_{k,m} - S_k^*) \cdot \prod_{\tau=m+1}^{N+1} (1 - \gamma_{k,\tau}) + \sum_{\tau=m+1}^{N} \left[ \prod_{s=\tau+1}^{N+1} (1 - \gamma_{k,s}) \right] \gamma_{k,\tau} \left( F_k(\boldsymbol{S}_{\tau-1}) - S_k^* + \bar{\xi}_{k,\tau} \right)$$
$$+ \gamma_{k,N+1} \left( F_k(\boldsymbol{S}_N) - S_k^* + \bar{\xi}_{k,N+1} \right).$$
$$(39)$$

Note that $\prod_{s=N+2}^{N+1} (1 - \gamma_{k,s}) = 1$, which implies the last term of (39) can be rewritten

$$\gamma_{k,N+1} \left( F_k(\boldsymbol{S}_N) - S_k^* + \bar{\xi}_{k,N+1} \right) = \prod_{s=N+2}^{N+1} (1 - \gamma_{k,s}) \cdot \gamma_{k,N+1} \left( F_k(\boldsymbol{S}_N) - S_k^* + \bar{\xi}_{k,N+1} \right).$$

So we can further rewrite the right hand side of (39) to

$$S_{k,N+1} - S_k^*$$
$$= (S_{k,m} - S_k^*) \cdot \prod_{\tau=m+1}^{N+1} (1 - \gamma_{k,\tau}) + \sum_{\tau=m+1}^{N} \left[ \prod_{s=\tau+1}^{N+1} (1 - \gamma_{k,s}) \right] \gamma_{k,\tau} \left( F_k(\boldsymbol{S}_{\tau-1}) - S_k^* + \bar{\xi}_{k,\tau} \right)$$
$$+ \prod_{s=N+2}^{N+1} (1 - \gamma_{k,s}) \cdot \gamma_{k,N+1} \left( F_k(\boldsymbol{S}_N) - S_k^* + \bar{\xi}_{k,N+1} \right)$$
$$= (S_{k,m} - S_k^*) \cdot \prod_{\tau=m+1}^{N+1} (1 - \gamma_{k,\tau}) + \sum_{\tau=m+1}^{N+1} \left[ \prod_{s=\tau+1}^{N+1} (1 - \gamma_{k,s}) \right] \gamma_{k,\tau} \left( F_k(\boldsymbol{S}_{\tau-1}) - S_k^* + \bar{\xi}_{k,\tau} \right).$$

Therefore, the statement holds for time $n = N + 1$, which completes the proof. $\qquad\square$

### A.8 PROOF OF LEMMA 6

*Proof of Lemma 6.* We prove this lemma by induction. When $n = m$, the statement is obviously true. Suppose it is true for time $n = N$, *i.e.,*

$$\prod_{\tau=m}^{N} (1 - \gamma_{k,\tau}) + \sum_{\tau=m}^{N} \left[ \prod_{s=\tau+1}^{N} (1 - \gamma_{k,s}) \right] \gamma_{k,\tau} = 1.$$

Then consider $n = N + 1$,

$$\prod_{\tau=m}^{N+1} (1 - \gamma_{k,\tau}) + \sum_{\tau=m}^{N+1} \left[ \prod_{s=\tau+1}^{N+1} (1 - \gamma_{k,s}) \right] \gamma_{k,\tau}$$

$$= (1 - \gamma_{k,N+1}) \cdot \prod_{\tau=m}^{N} (1 - \gamma_{k,\tau}) + \sum_{\tau=m}^{N} \left[ \prod_{s=\tau+1}^{N+1} (1 - \gamma_{k,s}) \right] \gamma_{k,\tau} + \gamma_{k,N+1}$$

$$= (1 - \gamma_{k,N+1}) \cdot \prod_{\tau=m}^{N} (1 - \gamma_{k,\tau}) + (1 - \gamma_{k,N+1}) \cdot \sum_{\tau=m}^{N} \left[ \prod_{s=\tau+1}^{N} (1 - \gamma_{k,s}) \right] \gamma_{k,\tau} + \gamma_{k,N+1}$$

$$= (1 - \gamma_{k,N+1}) \left[ \prod_{\tau=m}^{N} (1 - \gamma_{k,\tau}) + \sum_{\tau=m}^{N} \left( \prod_{s=\tau+1}^{N} (1 - \gamma_{k,s}) \right) \gamma_{k,\tau} \right] + \gamma_{k,N+1}$$

$$= 1,$$

where the first equation is from $\left[ \prod_{s=\tau+1}^{N+1} (1 - \gamma_{k,s}) \right] \gamma_{k,\tau} = \gamma_{k,N+1}$ when $\tau = N + 1$, and the last equality is obtained from the induction assumption for $n = N$. Therefore, the statement holds for time $n = N + 1$, which completes the proof. $\square$

### A.9 PROOF OF LEMMA 7

*Proof of Lemma 7.* Consider

$$\log \left[ \prod_{\tau=1}^{n} (1 - \gamma_{k,\tau}) \right] = \sum_{\tau=1}^{n} \log (1 - \gamma_{k,\tau}) \leq - \sum_{\tau=1}^{n} \gamma_{k,\tau},$$

where the inequality is due to $\log x \leq x - 1$ for all $x > 0$. We know that $\sum_{n=1}^{+\infty} \gamma_{k,n} = \infty$ almost surely for all $k = 1, \cdots, 8$ from Lemma 2. Therefore,

$$\lim_{n \to \infty} \log \left[ \prod_{\tau=1}^{n} (1 - \gamma_{k,\tau}) \right] = -\infty, \quad a.s.$$

which implies $\prod_{\tau=1}^{\infty} (1 - \gamma_{k,\tau}) = 0$ almost surely. $\square$

