# OpenReview forum: "No Algorithmic Collusion in Two-Player Blindfolded Games with Thompson Sampling"
_ICLR.cc/2025/Conference — Submitted to ICLR 2025_

### Official Review · Reviewer_7fhH · 2024-10-30

**Soundness:** 3
**Presentation:** 3
**Contribution:** 3
**Rating:** 5
**Confidence:** 3

**Summary:**

This paper presents a negative result in the area of repeated games and learning.  They study a scenario to two players, each having two actions.  The players are "blindfolded" in that they only observe past actions and payoffs of themselves.  The main result shows that if the players both use Thompson sampling, then then two players convers to the unique Nash equilibrium under mild conditions, i.e., no collusion arises.  Proving this result requires new technical ideas beyond the typical approaches in the literature due to the lack of simultaneous updates and the lack of global Lipshitzness.

**Strengths:**

Learning in games is a thriving area and understanding when agents converge or not to different forms of equilibria is timely and interesting.  In particular, quantifying what factors lead to collusion (or the lack of it) is a central question in the literature.

The paper is clearly written and the authors do a solid job of placing their results in the literature.

The inclusion of numerical experiments in what is primarily and theoretical paper is appreciated.

**Weaknesses:**

The paper considers a very narrow setting - two-players, two-actions with both using a very specific algorithm.  The contribution of the paper would be larger if if could state a broader class of models/algorithms also maintained "no collusion".  As written, the result feels more like a curiosity worth investigating further than a major contribution, i.e., it shows a property of an example without exploring the limits of where the property holds.   Example are, of course important, but the contribution would be larger if the setting could be expanded to probe the limits by considering, e.g., more players, more actions, other learning rules, etc.

A particular limitation of interest to generalize is the specific assumption of Thompson sampling as the algorithm.  Studying a broader class of algorithms would increase the level of contribution significantly.

Table 1 highlights some challenges as compared to "classic" approaches in the literature.  However, there are a wide variety of modern approaches proposed in the past few years as this literature has exploded.  The paper does not highlight the challenges with applying these modern analyses (many of which consider more general settings) to the current results in the paper.  As starting point would be to highlight the challenges associated with applying the techniques in the papers mentioned in the last paragraph of Section 1.  See the question below.

**Questions:**

How much more generality in the model could your proposed techniques handle?  What generalizations to the model will require new techniques? What more general class of algorithms could your techniques apply to?

What are the technical contributions in your analysis relative to more recent approaches than those summarized in Table 1.  For example, those mentioned in the last paragraph of Section 1.

---

> ### Author Response · Authors · 2024-11-23
> **Official Comment by Authors**
>
> We appreciate your careful reading of the paper and constructive criticism. Below we address your major concerns.
>
> - **Specific models and algorithms.** We acknowledge that the two-player two-action setting is somewhat restricted. We would like to point out that because of the technical challenge of the problem, even this setting requires a very delicate analysis, as highlighted in the appendix of the proof. We view the analysis as one of the major contributions of the paper. We are working on relaxing the assumption, and the convergence might still hold when there are *more than two actions.*
> However, we do not think that studying Thompson sampling is a limitation of the work. The recent focus of algorithmic collusion has been on specific algorithms, such as Q-learning [1] and UCB [2]. Note that both works use numerical simulations to show that Q-learning and UCB are *susceptible to* algorithmic collusion. In this work, we have shown that Thompson sampling will not collude. It is clear that the analysis has to be carried out for individual algorithms case by case. The same technique (stochastic approximation) would not work for deterministic or more discrete algorithms such as UCB or $\\epsilon$-greedy.
> To answer your first set of questions more specifically, we do see stochastic approximation and the sample-pathwise analysis a potentially powerful tool when analyzing randomized online learning algorithms in the game setting, such as Boltzmann Exploration. But the analysis will be very different, requiring different sets of assumptions and techniques. This is similar to their regret analysis typically requiring individual treatments.
>
> - **Recent papers and modern approaches.**  While there have been many recent studies on stochastic approximation (SA) (see the papers mentioned in the last paragraph of Section 1), they have primarily focused on finite-time analysis of SA, compared with the classical literature which focused on asymptotic convergence of SA. None of these recent approaches/results on finite time analysis can be applied to our problem. Note that the recent advance is to push the existing asymptotic analysis to the non-asymptotic regime, while the challenge arising from our setting is regarding the asymptotic analysis itself. For example, Haque et al. (2023) study finite-time analysis of two-time scale SA, but our SA scheme is not a two-time scale SA (because the posterior of the inferior action is only updated infrequently and sporadically). Similarly, Qu & Wierman (2020) study finite-time analysis of asynchronous SA, but our SA scheme is also different from their asynchronous SA (because the updating frequencies of different actions  are not of the same order). Therefore, we can not apply these modern approaches in the recent SA literature. We believe that our asymptotic convergence analysis of the SA scheme that arises in the game setting with Thompson sampling is novel, and this is one of our main technical contributions.
>
> - **Recent papers on learning in games.** The recent studies on the convergence of learning algorithms in games have been focusing on games with continuous actions and/or gradient information [3]. Moreover, the payoff function typically has convexity assumptions. The analysis cannot be applied to Thompson sampling because the action set is discrete and it doesn’t use gradient-based approaches.
>
> [1] Calvano, Emilio, et al. "Artificial intelligence, algorithmic pricing, and collusion." *American Economic Review* 110.10 (2020): 3267-3297.
> [2] Hansen, Karsten T., Kanishka Misra, and Mallesh M. Pai. "Frontiers: Algorithmic collusion: Supra-competitive prices via independent algorithms." *Marketing Science* 40.1 (2021): 1-12.
> [3] Mertikopoulos, Panayotis, and Zhengyuan Zhou. "Learning in games with continuous action sets and unknown payoff functions." *Mathematical Programming* 173 (2019): 465-507.

---

> > ### Comment · Reviewer_7fhH · 2024-11-25
> >
> > Thank you for the thoughtful response.  I have spent the weekend looking into the proofs in more detail and trying to understand the challenges more concretely.  I now understand more where the novelty in the analysis falls and appreciate the technical work more deeply.   I will raise my score to indicate this.
> >
> > A small additional suggestion is to discuss connections/contrasts to the literature on no-swap regret algorithms in the learning in games literature, such as https://arxiv.org/abs/2402.09549 (and the references therein).  The results are different but share a similar motivation/perspective.

---

> ### Author Response · Authors · 2024-11-27
>
> Thank you for the careful reading of the paper and improving the score. We reviewed a few papers on swap regret and learning. They are indeed related. We know that an algorithm that minimizes internal regret will converge to the Nash equilibrium. Swap regret is an extension of internal regret which implies the same convergence to NE. One stream of works discuss the reduction from swap regret to external regret, e.g., [1,2]. They focus on how to achieve low internal/swap regret given a low external regret algorithm. Another type of papers present algorithms that have low swap regret [3, 4]. In addition, [5] shows the relationships between no-regret (i.e. no-external-regret), no-swap-regret and Pareto-optimal algorithms. We will add these studies to our literature review.
>
> We believe the motivation and analysis of this study is different. The primary reason is that Thompson sampling does not have sublinear (internal/swap) regret in the adversarial setting. Thus, the results in the literature do not imply no collusion.
>
> [1] Blum, A., & Mansour, Y. (2007). From external to internal regret. Journal of Machine Learning Research, 8(6).\
> [2] Dagan, Y., Daskalakis, C., Fishelson, M., & Golowich, N. (2024, June). From External to Swap Regret 2.0: An Efficient Reduction for Large Action Spaces. In Proceedings of the 56th Annual ACM Symposium on Theory of Computing (pp. 1216-1222).\
> [3] Anagnostides, I., Farina, G., Kroer, C., Lee, C. W., Luo, H., & Sandholm, T. (2022). Uncoupled learning dynamics with $ o (\log t) $ swap regret in multiplayer games. Advances in Neural Information Processing Systems, 35, 3292-3304.\
> [4] Peng, B., & Rubinstein, A. (2024, June). Fast swap regret minimization and applications to approximate correlated equilibria. In Proceedings of the 56th Annual ACM Symposium on Theory of Computing (pp. 1223-1234).\
> [5] Arunachaleswaran, E. R., Collina, N., & Schneider, J. (2024). Pareto-Optimal Algorithms for Learning in Games. arXiv preprint arXiv:2402.09549.

---

> > ### Comment · Reviewer_7fhH · 2024-11-27
> >
> > I'm glad to see you went deeply into that line of work.  Thank you.  Indeed, I agree that the line of work is considering a different but related problem that will be good to highlight in the literature review.

---

### Official Review · Reviewer_h5co · 2024-11-06

**Soundness:** 4
**Presentation:** 3
**Contribution:** 4
**Rating:** 8
**Confidence:** 3

**Summary:**

This work studies the dynamics of a repeated game where the players play using Thompson sampling based on the realized payoffs of their actions. This is referred to as a "blindfolded" game since the players do not get to see the impact of the other players' actions, or even model the other players, and play purely based on the payoffs of the actions (which are influenced by the actions chosen by the other players). Thompson sampling is an algorithm with a no-regret guarantee in multi-armed bandits problems with a stationary distribution for each arm, which is not necessarily the case in the scenario of blindfolded games, due to the adaptive and non-stationary behavior of other players. The work establishes that in two-player, two-action games satisfying some conditions including the existence of a unique pure strategy Nash Equilibrium (PSNE), the dynamic of play converge in the last iterate to this PSNE. The work is situated in the context of previous papers studying algorithmic collusion in repeated pricing games where price setters use reinforcement learning to set prices and end up converging to collusive, non-equilibrium strategy profiles. In particular, this result is framed as a counterpoint to theoretical analysis of Hansen et. al. showing UCB algorithms (another class of no-regret algorithms for multi-armed bandits) can converge to collusive strategy profiles.

**Strengths:**

The main strength of this paper is in studying the intricate dynamics of agents employing Thompson sampling against each other.  Unlike no-regret algorithms such as hedge/ multiplicative weights, Thompson sampling is a mis-specified algorithms in the context of games, and there is a notable lack of tools to analyze the dynamics they induce. This result is a non-trivial first contribution towards understanding these dynamics. The paper contains some interesting technical tools, including setting up a natural state space to track the dynamics and uses a sample-path-wise approach, building upon tools from Tsitsiklis 1994 to prove the result. Even though the result only holds for under some fairly restrictive conditions, it sets up some interesting open problems for future research to pursue.

**Weaknesses:**

The main weakness of the result is the set of restrictive conditions on the class of games, such as only two actions, a unique pure strategy Nash Equilibrium and assumptions about the payoff of the Nash Equilibrium not being much worse than the other action profiles. In particular, it is not clear if these even includes the simple pricing games that related work, such as Hansen et. al., Calvano et. al. study. These pricing games offer a compelling reason to study the emergence of collusion in the dynamics. Additionally, it is clear that results such as the folk theorem show the existence of algorithmic strategies (however artificial) that induce collusion in these pricing games, making it interesting to study if particular algorithmic strategies result in dynamics leading to collusion. It would help if the work could explain why these games may be interesting and discuss if collusion is possible under different algorithmic strategies employed by the players. The other weakness is the lack of discussion about how different parts of the proof slot together and exactly why Thompson sampling, as opposed to UCB/ other exploration-exploitation algorithms, provably converges to the Nash.

**Questions:**

Why is this class of games interesting? How does it compare the pricing games studied by related work? Is collusion possible at all using different algorithms for the players? Are any of the conditions absolutely necessary? Are there counterexamples showing the lack of one of these conditions breaks the result - one candidate that stands out in particular is the assumption about a unique PSNE.

---

> ### Author Response · Authors · 2024-11-23
> **Official Comment by Authors**
>
> Thank you for your positive assessment and constructive comments for our paper. Please see our response below.
>
> - **Assumptions.** We acknowledge that some assumptions do appear restrictive, especially Assumption 2\. From the numerical examples, the convergence may still hold in the absence of the assumption. We are currently pushing the analysis to see if it can be adapted to the more general case. However, the unique Nash equilibrium assumption is necessary, as demonstrated by the numerical experiment attached. In the literature it is also assumed when the action space is continuous and monotone/concave payoff functions [1,2].
>
> - **Pricing game.** Let us consider the following pricing game. There are two firms and each firm chooses between two prices, $p_L=0.6$, $p_H=0.7$. The linear demand function is $d_i(p_i,p_{-i})=0.5-p_i+0.4*p_{-i}$. We can calculate the following payoff results, $\\pi_{LL}=0.084, \\pi_{LH}=0.108, \\pi_{HL}=0.028, \\pi_{HH}=0.056$. The Nash equilibrium is $(L, L)$, and the corresponding payoff matrices satisfy our two assumptions. It shows that our problem setting could cover the simple pricing game.
>
> - **Results of different algorithmic strategies.** **(1) Why use Thompson sampling for both players.** This is a great point. Based on the analysis of Thompson sampling in multi-armed bandits, it has the following property: if the other player adopts a stationary policy, then the player using Thompson sampling will converge to the best response with $\\sqrt{n}$ regret. This property provides justification for using and analyzing Thompson sampling in a game. Moreover, most papers on algorithmic collusion focus on a single class of policies of the players, such as UCB [Hansen et. al.] or Q-learning [Calvano et. al.], and are based on simulations. We provide a theoretical analysis for Thompson sampling, which is arguably one of the most well-known learning algorithms. We will add the remarks to the revision. **(2) UCB versus Thompson sampling.** The reason why UCB does not converge to the NE is due to the lack of randomness. In fact, the dynamic of the game under UCB as a deterministic dynamical system depends on the initial state, which results in non-convergence. The internal randomness in Thompson sampling helps the convergence, as shown by our analysis of the stochastic approximation. But this is not the only reason and the convergence has something to do with the steps of Thompson sampling specifically. For example, [3] show that randomized algorithms such as $\\epsilon$-greedy may still lead to collusion (does not converge to NE). We will remark on this in the revision.
>
> - **Counterexamples without these assumptions.** As you pointed out, we show that with mixed-strategy NEs, the game may converge to one of the pure-strategy NEs probabilistically, or not converge in new experiments (please see Section A.1 in the Appendix of the revision). The experiments show the necessity of a unique pure-strategy NE.
>
> [1] Cesa-Bianchi, Nicolo, and Gábor Lugosi. *Prediction, learning, and games*. Cambridge university press, 2006.
> [2] Jordan, Michael, Tianyi Lin, and Zhengyuan Zhou. "Adaptive, doubly optimal no-regret learning in strongly monotone and exp-concave games with gradient feedback." *Operations Research* (2024).
> [3] Klein, Timo. "Autonomous algorithmic collusion: Q‐learning under sequential pricing." *The RAND Journal of Economics* 52.3 (2021): 538-558.

---

> > ### Comment · Reviewer_h5co · 2024-11-25
> >
> > Thanks for responding to the review.
> >
> > The contrast between Thompson sampling and other randomized algorithms is well made -- this adds value to the bespoke analysis demonstrated by your work for Thompson sampling.
> >
> > I'm also satisfied with your explanation about the need for unique PSNE, and agree that it is a reasonable assumption.

---

> > > ### Author Response · Authors · 2024-11-27
> > >
> > > Thank you for your feedback. We appreciate your earlier comment and are pleased that our response addressed it.

---

### Official Review · Reviewer_fWSu · 2024-11-06

**Soundness:** 3
**Presentation:** 2
**Contribution:** 2
**Rating:** 6
**Confidence:** 3

**Summary:**

The authors study repeated play in two-player, two-action games that admit a unique pure Nash equilibrium. They assume a minimal information setting. In particular, across all rounds, players only observe their own payoffs. In addition, the player's payoffs are subjected to zero-mean normally distributed noise.

Their main result, Theorem 1, states that under Assumptions 1 and 2, if both players assume a Bayesian point of view and update some particular prior distributions based on Thompson sampling, then almost surely they will converge to the unique Nash equilibrium of the game. The authors' ultimate claim is that under these assumptions algorithmic collusion is almost surely not possible.

**Strengths:**

1. The information setting studied by the authors is quite realistic.
2. The analysis, to the best of my knowledge, is correct.

**Weaknesses:**

1. Assumption 1 is indeed mild. However, Assumption 2 seems to be quite strong for the study of algorithmic collusion. In particular, under Assumption 2, the game's unique Nash equilibrium cannot be, as the authors point out, much worse than any other outcome. Therefore, independently of the algorithm used, gains from an algorithmic collusion can be minimal at best. Would the authors provide some further analysis of the implications of this assumption towards this particular claim?
2. A possible weakness is also the implicit assumption that the game's unique Nash equilibrium is pure. The uniqueness of the game's Nash equilibrium is a reasonable assumption for the study of the algorithm's convergence. However, if the game doesn't have a special structure, e.g., being a potential game, this equilibrium might not be pure. Could the authors further motivate this assumption?

**Questions:**

Kindly refer to the weaknesses.

---

> ### Author Response · Authors · 2024-11-23
> **Official Comment by Authors**
>
> We thank you for your constructive comments. We provide responses to your comments and questions below.
>
> - **Implication of Assumption 2.** Assumption 2 indeed seems strong and there is some room for improvement. In particular, the technical assumption on the payoff matrix is not necessary: an example of the convergence in the absence of Assumption 2 is demonstrated in the numerical experiment. However, we want to emphasize that the technical complexities of the problem necessitate a nuanced analysis, as demonstrated in the appendix of the proof. Currently, we do not know how to relax Assumption 2 and adapt the analysis, but it is a very important future direction we are pushing.
>
> - **Mixed Nash equilibrium.** We argue that compared to the literature, the assumption on a unique NE is not too strong. When studying the convergence to the NE of games with continuous action spaces, many studies assume a unique pure-strategy NE and monotone/concave payoff functions [1,2]. In new experiments (please see Section A.1 in the Appendix of the revision), we show that with mixed-strategy NEs, the game may converge to one of the pure-strategy NEs probabilistically, or not converge. The experiments show the necessity of a unique pure-strategy NE.
>
>
> [1] Cesa-Bianchi, Nicolo, and Gábor Lugosi. *Prediction, learning, and games*. Cambridge university press, 2006.
> [2] Jordan, Michael, Tianyi Lin, and Zhengyuan Zhou. "Adaptive, doubly optimal no-regret learning in strongly monotone and exp-concave games with gradient feedback." *Operations Research* (2024).

---

### Official Review · Reviewer_pH4Y · 2024-11-06

**Soundness:** 2
**Presentation:** 2
**Contribution:** 2
**Rating:** 5
**Confidence:** 4

**Summary:**

This paper mainly focuses on dynamics in two-player blindfolded games, where both players follow Thompson sampling to choose their actions.
Under some assumptions, the author demonstrates that the game dynamics converge to the pure Nash equilibrium.
The proof utilizes the fact that the game dynamics can be viewed as a special form of stochastic approximation.
Furthermore, the author also experimentally shows that each player's strategy converges to a pure Nash equilibrium in some games.

**Strengths:**

* The problem is well-motivated. Analyzing the celebrated MAB algorithm in games is of great importance to the research community.
* It seems novel to provide last-iterate convergence guarantees for Thompson sampling in online learning in games.

**Weaknesses:**

My primary concerns are centered on the assumptions on two-player games.
Firstly, the last-iterate convergence results are only provided for **two-action** games.
The analysis heavily depends on the fact that, in the two-action games, the action probability can be formulated in a closed-form expression (as in Eq. (7)).
However, I am curious if similar expressions can be derived for more general games.
Secondly, the assumption regarding the existence and uniqueness of the pure Nash equilibrium appears to be strong.
I am uncertain about the practical applications of games with these assumptions.
If these assumptions limit the practical application of this study, it would be advantageous to provide any convergence or divergent results in games that do not adhere to these assumptions, such as two-player zero-sum games.

Moreover, a recent study [1] primarily provided last-iterate convergence rates in two-player zero-sum normal-form games and Markov games under bandit feedback.
I believe the bandit feedback setting closely resembles the blindfolded setting.
Hence, the relationship with [1] should be clarified.

[1] Cai, Y., Luo, H., Wei, C.-Y., and Zheng, W. Uncoupled and convergent learning in two-player zero-sum Markov games. NeurIPS, 2023.

**Questions:**

My main concerns and questions are outlined in Weaknesses.

---

> ### Author Response · Authors · 2024-11-23
> **Official Comment by Authors**
>
> Thank you for your constructive criticism of the work. Please see the response below.
>
> - **Assumptions on two actions.** We recognize that the two-player, two-action model has its limitations. Nonetheless, it's important to highlight that the difficulty of this problem makes it challenging to analyze even the simplest problem configurations. Note that in our proof, we do not use the closed-form expressions of the normal distribution (in fact, the CDFs do not have a closed form). Instead, we rely on some analytical properties of normal distributions, such as the scaling with number of plays and the decay rate of the tail probability. These properties tend to hold for more than two actions. Therefore, it is likely that the analysis can be generalized to more than two actions and we are working on it.
>
> - **Assumptions on pure-strategy NE.** Regarding your comment on the assumption of a unique pure-strategy NE, we argue that compared to the literature, this assumption is not too strong. When studying the convergence to the NE of games with continuous action spaces, many studies assume a unique pure-strategy NE and monotone/concave payoff functions \[1,2\]. Moreover, in the new experiments (please see Section A.1 in the Appendix of the revision), we show that with mixed-strategy NEs, the game may converge to one of the pure-strategy NEs probabilistically, or not converge. The experiments show the necessity of the assumption.
>
> - **Practical applications.** Let us consider the following pricing game. There are two firms and each firm chooses between two prices, $p_L=0.6$, $p_H=0.7$. The linear demand function is $d_i(p_i,p_{-i})=0.5-p_i+0.4*p_{-i}$. We can calculate the following payoff results, $\\pi_{LL}=0.084，\\pi_{LH}=0.108, \\pi_{HL}=0.028, \\pi_{HH}=0.056$. The Nash equilibrium is $(L, L)$, and the corresponding payoff matrices satisfy our two assumptions. Therefore, there are practical applications that could meet these assumptions.
>
> - **Comparison with Cai et al (2023).** Thank you for bringing this paper to our attention and we regret not including it in our initial submission. It is indeed an important paper. Upon reading the paper more carefully, we believe the motivation and analysis of our study deviates from Cai et al (2023). In their paper, the goal is to provide an algorithm that can converge to the Nash equilibrium with non-asymptotic analyses, based on EXP3. It is probably not surprising given that algorithms like EXP3 which minimize the internal regret have been shown to converge to the Nash equilibrium. Their study extends this notion to more general settings such as the last-iterate convergence, which is remarkable. In our setting, we focus on a specific algorithm, Thompson sampling, which is widely used in practice, and investigate its asymptotic property in a game. Therefore, the motivation (design of an algorithm versus analysis of an existing algorithm) differs. Moreover, the analysis of Thompson sampling in this study deviates significantly from the analysis of EXP3-type algorithms proposed in Cai et al (2023).
>
> [1] Cesa-Bianchi, Nicolo, and Gábor Lugosi. *Prediction, learning, and games*. Cambridge university press, 2006.
> [2] Jordan, Michael, Tianyi Lin, and Zhengyuan Zhou. "Adaptive, doubly optimal no-regret learning in strongly monotone and exp-concave games with gradient feedback." *Operations Research* (2024).

---

> > ### Comment · Reviewer_pH4Y · 2024-11-27
> >
> > Thank you for your response and detailed explanations.
> >
> > I acknowledge that this work's motivation differs from that of Cai et al. (2023), and the theoretical analysis of Thomson sampling presents a degree of novelty.
> > However, I remain curious about the feasibility of extending the theoretical results from two-action games to a wider range of games.
> > Additionally, the pricing game example provided by the author seems to lack practicality since it only has two prices.
> > It would be beneficial to discuss the potential applicability of the derived theoretical results to more general pricing games (e.g., a pricing game where more than three prices can be selected).

---

> ### Author Response · Authors · 2024-11-27
>
> Thank you for your constructive comment.
>
> On your comment about the pricing game: indeed we are only considering two prices. That is because the number of prices the players can charge is the number of actions in the game. Since our work focuses on two actions, we use two prices in the pricing game.
>
> On whether we can extend it to multiple actions: we are working on this part in the last few days. The main technical challenge is two-fold. First, because of the nature of the sample-path-wise argument, we need to track the dynamic of the state. For more than two actions, the state space increases and the notation becomes heavy. We are trying to figure out a compact vectorized representation to replicate the analysis. Second, the key of the analysis is to characterize the region in the state space where the dynamic is *not Lipschitz continuous* asymptotically. For two actions, it is where the empirical averages of the two actions are equal, i.e., $x_1=x_2$. For multiple actions, it is where $x_i=\max\\{x1,x2,...,x_k\\}$ for all $i$. It is a region that is a lot harder to analyze. Since we need to show the dynamics will escape the region, the analysis is more complicated for multiple actions. We are still working to extend the analysis.

---

### Meta-Review · Area_Chair_E2dn · 2024-12-19

**Metareview:**

This paper studies learning with bandit feedback - that is, observing only one's own realized payoffs - in $2\times2$ normal form games with a unique, strict Nash equilibrium. Both players are assumed to employ a Thompson sampling algorithm and the authors show that, under the above assumptions, the learning process converges to equlibrium (what the authors call "no collusion").

Most of the reviewers were on the fence about this paper, for several reasons:
- First, the paper concerns only $2\times2$ games. Even in the bandit setting - which the authors call the "blindfolded game" - there is a wide array of methods in the literature for achieving convergence to Nash equilibrium. Admittedly, the use of Thompson sampling by both players has not been studied extensively in the literature, but the problem of learning with bandit feedback in small games otherwise has a vast literature (which the authors do not cover / seem to be aware of).
- The reviews also raised concerns regarding the assumption that the game has a unique pure Nash equilibrium. Since the authors also assume that there are no payoff ties in the game, this means that the game admits a unique *strict* Nash equilibrium. In turn, since the paper only concerns $2\times2$ games, this means that the game is strategically equivalent to the Prisoner's Dilemma, for which there is an even wider array of algorithms converging to Nash equilibrium, even with bandit feedback (for example, any algorithm that eliminates dominated strategies - like EXP3 or any bandit variant of FTRL - would suffice).

The paper was not championed during the discussion phase and it was ultimately decided to make a "reject" recommendation.

**Additional Comments On Reviewer Discussion:**

In view of the limitations mentioned above, it was not possible to make the case that the paper clears the bar for acceptance. The paper was not championed by any of the reviewers during the discussion phase, and the more reviewers were not swayed by the authors' replies, so the conclusion was that the paper cannot be accepted at this point.

---

### Decision · Program_Chairs · 2025-01-22

Reject